# Consistent global structures of complex RNA states through multidimensional chemical mapping

Clarence Yu Cheng[1], Fang-Chieh Chou[1], Wipapat Kladwang[1], Siqi Tian[1], Pablo Cordero[2], Rhiju Das[1,2,3]*

[1]Department of Biochemistry, Stanford University, Stanford, United States; [2]Biomedical Informatics Program, Stanford University, Stanford, United States; [3]Department of Physics, Stanford University, Stanford, United States

**Abstract** Accelerating discoveries of non-coding RNA (ncRNA) in myriad biological processes pose major challenges to structural and functional analysis. Despite progress in secondary structure modeling, high-throughput methods have generally failed to determine ncRNA tertiary structures, even at the 1-nm resolution that enables visualization of how helices and functional motifs are positioned in three dimensions. We report that integrating a new method called MOHCA-seq (Multiplexed •OH Cleavage Analysis with paired-end sequencing) with mutate-and-map secondary structure inference guides Rosetta 3D modeling to consistent 1-nm accuracy for intricately folded ncRNAs with lengths up to 188 nucleotides, including a blind RNA-puzzle challenge, the lariat-capping ribozyme. This multidimensional chemical mapping (MCM) pipeline resolves unexpected tertiary proximities for cyclic-di-GMP, glycine, and adenosylcobalamin riboswitch aptamers without their ligands and a loose structure for the recently discovered human HoxA9D internal ribosome entry site regulon. MCM offers a sequencing-based route to uncovering ncRNA 3D structure, applicable to functionally important but potentially heterogeneous states.

*For correspondence: rhiju@stanford.edu

Competing interests: The authors declare that no competing interests exist.

## Introduction

RNAs fold into and interconvert between specific secondary and tertiary structures to perform a wealth of essential biological functions (*Atkins et al., 2011*; *Breaker, 2012*). In particular, medium-sized non-coding RNA (ncRNA) domains in the 100- to 300-nucleotide size range have been discovered to form specific tertiary structures for sensing intra-cellular metabolites, regulating gene expression, and catalyzing precise splicing events (*Smith et al., 2009*; *Butler et al., 2011*; *Peselis & Serganov, 2012*; *Meyer et al., 2014*). Similarly functional behaviors may be carried out by the tens of thousands of ncRNA domains expressed in complex organisms (*Amaral et al., 2008*). High-resolution structure determination techniques such as crystallography, NMR, and cryo-electron microscopy are providing critical insights into ncRNA behavior but are challenged by the large sizes, conformational heterogeneity, and multiple functional states of most biological ncRNA domains (*Weeks, 2010*). As an alternative methodology, there is a rich history of specialized biochemical interrogation, phylogenetic analysis, and manual model building to achieve global 3D structures of ncRNAs, including the first discovered ribozymes (*Kim and Cech, 1987*; *Lehnert et al., 1996*; *Bergman et al., 2004*; *Kazantsev and Pace, 2006*; *Lipfert et al., 2008*). These models have reached nanometer accuracies, comparable to the spacing between adjacent nucleotides (6 Å) and significantly smaller than the diameter of an RNA helix (23 Å). While not atomic accuracy, such models have guided visualization of how an ncRNA's motifs and helices interact to effect functions like catalysis and ligand binding, fueling powerful experiments and evolutionary insights (*Hougland et al., 2006*). However, the experimental

**eLife digest** Our genetic material, in the form of molecules of DNA, provides instructions for many different processes in our cells. To issue these instructions, particular sections of DNA are copied to make a type of molecule called ribonucleic acid (RNA). Some of these RNA molecules contain instructions to make proteins, but others—known as non-coding RNAs—regulate the activity of genes in cells.

The genetic information within RNA is encoded by the sequence of four different chemical parts called 'nucleotides'. RNA can exist as a single strand of nucleotides, but the nucleotides can also pair up in specific combinations to form sections of double-stranded RNA. Therefore, a single strand of non-coding RNA can fold into a complex three-dimensional shape that contains loops, twists, and bulges.

The three-dimensional structures of non-coding RNAs are crucial for their roles in cells, but the variety and complexity of shapes that they can form makes it technically difficult to study them. In 2008, researchers developed a new method called MOHCA that can map the positions of nucleotides that are close together in the three-dimensional structure. Highly reactive chemicals are attached to the nucleotides and these can react with, and damage, other nearby nucleotides. By detecting which nucleotides have been damaged, it is possible to map the positions of these nucleotides and decipher the structure of the RNA molecule using computer algorithms.

MOHCA is a promising approach, but the initial methods to find the damaged nucleotides were tedious and required specialized equipment. Now, Cheng, Das et al.—including some of the researchers involved in the 2008 work—have developed an improved version of MOHCA that uses readily available RNA sequencing techniques to find the damaged nucleotides. The RNA sequencing data are then analyzed by a new algorithm in the Rosetta computer modeling software.

Cheng, Das et al. used this newly developed 'MOHCA-seq' and Rosetta to reveal the structures of a human non-coding RNA and several other non-coding RNA molecules to a much higher level of detail than before. Together, MOHCA-seq and Rosetta provide a rapid method for researchers to decipher the three-dimensional structure of non-coding RNAs. This method is likely to speed up the analysis of the complex structures of non-coding RNAs. It will be useful in future efforts to work out what roles these RNAs play in cells, including their activity in cancer, neurodegeneration, and other diseases.

and modeling methods underlying these classic, medium-resolution approaches have not been fully automated or accelerated.

Chemical mapping techniques, such as hydroxyl radical footprinting and SHAPE, (*Weeks, 2010*) combined with new computational tools (*Cheng et al., 2015*), offer a promising route for expanding medium-resolution 3D modeling to ncRNA sequences and states that remain uncharacterized and for eventually refining such models to high resolution using computation or focused experiments (*Fleishman and Baker, 2012*). Chemical mapping data report on the solvent exposure or base-pairing propensity of each nucleotide and can guide ncRNA structure inference with computational modeling algorithms (*Das et al., 2008*; *Ding et al., 2012*; *Homan et al., 2014*). Recent advances have enhanced the throughput of chemical mapping analysis and the accuracy of ncRNA secondary structure determination, with high accuracy achieved by 'two-dimensional' methods that integrate systematic mutagenesis with high-throughput mapping (mutate-and-map, or M$^2$), analogous to multidimensional NMR spectroscopy but at nucleotide length scales (*Kladwang et al., 2011a*; *Yoon et al., 2011*; *Ding et al., 2012*; *Mortimer et al., 2012*; *Karabiber et al., 2013*; *Seetin et al., 2014*; *Siegfried et al., 2014*). However, barriers remain in 3D modeling, even at the modest 1-nanometer resolution required to visualize helix-level structures, to assess structural heterogeneity, or to make hypotheses for ligand-binding sites or tertiary interactions that can be experimentally tested.

The state-of-the-art in ncRNA 3D structure modeling has been rigorously established by the 'RNA-puzzles' trials (*Cruz et al., 2012*; *Miao et al., 2015*). These community-wide exercises provide stringent tests of available computational methods by challenging modelers to predict structures of ncRNA sequences that have been crystallized but whose structures have not been publicly released. The dissemination of SHAPE, DMS, hydroxyl radical footprinting, and M$^2$ measurements for recent

RNA-puzzles has enabled our and other groups to consistently achieve high-secondary structure prediction accuracies. Furthermore, computational integration of available data with the fragment assembly of RNA (FARNA) algorithm in Rosetta (*Cheng et al., 2015*) has led to tertiary structure models with accuracies of 1 nanometer or better (all-heavy-atom root-mean-squared-deviation, RMSD). Unfortunately, these computational methods still generate alternative 3D structures consistent with the available data. Out of the 5 to 10 different structures submitted by each participating modeling team for each puzzle, we and others have been unable to rank the models with correct tertiary helical arrangements as our top submissions, much less confidently assign accuracies to these models or select candidates for computationally expensive refinement. We report herein an experimental/computational advance that resolves this barrier of 1-nm resolution global structure determination.

While $M^2$ efficiently infers Watson–Crick base pairs, the RNA-puzzles trials have exposed a critical need for rapid experimental methods to comprehensively determine nucleotide pairs that are not in direct atom–atom contact but are brought into nanometer-scale proximity by an ncRNA's tertiary structure. Such a method would be a nucleotide-resolution analog to nuclear Overhauser effect (NOE) NMR spectroscopy, which detects 'through-space' proximities between atoms that are not necessarily bonded. This problem prompted us to revisit a technique for discovering RNA tertiary proximities termed Multiplexed •OH (hydroxyl radical) Cleavage Analysis (MOHCA) (*Das et al., 2008*), which reports on the cleavage of nucleotides by hydroxyl radicals generated at sources 10–30 Å away that are tethered to other nucleotides. Unfortunately, the original protocol (renamed 'MOHCA-gel' herein) required specially synthesized 2′-NH$_2$-2′-deoxy-α-thio-nucleotide triphosphates for random radical source introduction and on-demand backbone scission at co-introduced phosphorothioates; a customized two-dimensional gel electrophoresis setup; radioactive labeling for the readout; and multiple experiments with both 5′ and 3′ end-labeling to achieve reasonable signal-to-noise (*Das et al., 2008*). These factors prevented MOHCA-gel from entering routine use or being tested along with $M^2$ in RNA-puzzles trials.

We hypothesized that the recent availability of tabletop deep sequencing would enable an accelerated and higher-precision 'MOHCA-seq' protocol and that these data would complement $M^2$ and Rosetta to give a multidimensional chemical mapping (MCM) pipeline for routine use in ncRNA 3D structure characterization. Here, we present the development, benchmarks, and a blind test of the MCM methodology, followed by applications to detect preformed structure in ligand-free riboswitch aptamer states and a recently discovered human ncRNA regulon.

## Results

### Deep proximity maps of ncRNA 3D structure from single experiments

To complement the prior $M^2$ method for secondary structure inference (*Figure 1A*) with tertiary proximity maps, we developed and tested the MOHCA-seq workflow, shown in *Figure 1B*. The workflow was designed to read out proximal nucleotide pairs by introducing radical sources at random locations into the RNA, creating radicals that initiate strand breaks at positions close in three dimensions to each source, and mapping the sequence positions of each strand break and its parent source. We randomly incorporated 2′-NH$_2$-2′-dATP during transcription so that each RNA contained on average one or fewer modifications and then coupled these sites to isothiocyanobenzyl-Fe(III)•EDTA. We chose these reagents because they are commercially available and therefore can be readily acquired by other laboratories. These RNAs were purified and then folded in buffer containing 50 mM Na-HEPES pH 8.0, 10 mM MgCl$_2$, and any ligands appropriate for the system (see below and 'Materials and methods'). After folding, RNAs were fragmented by activating the Fenton reaction at the backbone-tethered Fe(III) atoms using ascorbate as a reducing agent (*Das et al., 2008*). In the prior MOHCA-gel approach, electrophoresis in one dimension separated RNAs at strand breaks, and then fragments were cleaved at the location of the radical source by iodine-catalyzed scission of phosphorothioate tags (incorporated with the 2′-NH$_2$ modifications), before electrophoresis in the second, perpendicular direction (*Das et al., 2008*). The resulting patterns gave information on pairs of proximal nucleotides based on their position in the 2D gel, but the method's low signal-to-noise necessitated tedious experiments with radical sources separately incorporated at A, C, G, and U residues, each carried out in triplicate, each with at least two different gel electrophoresis times, to achieve a proximity map.

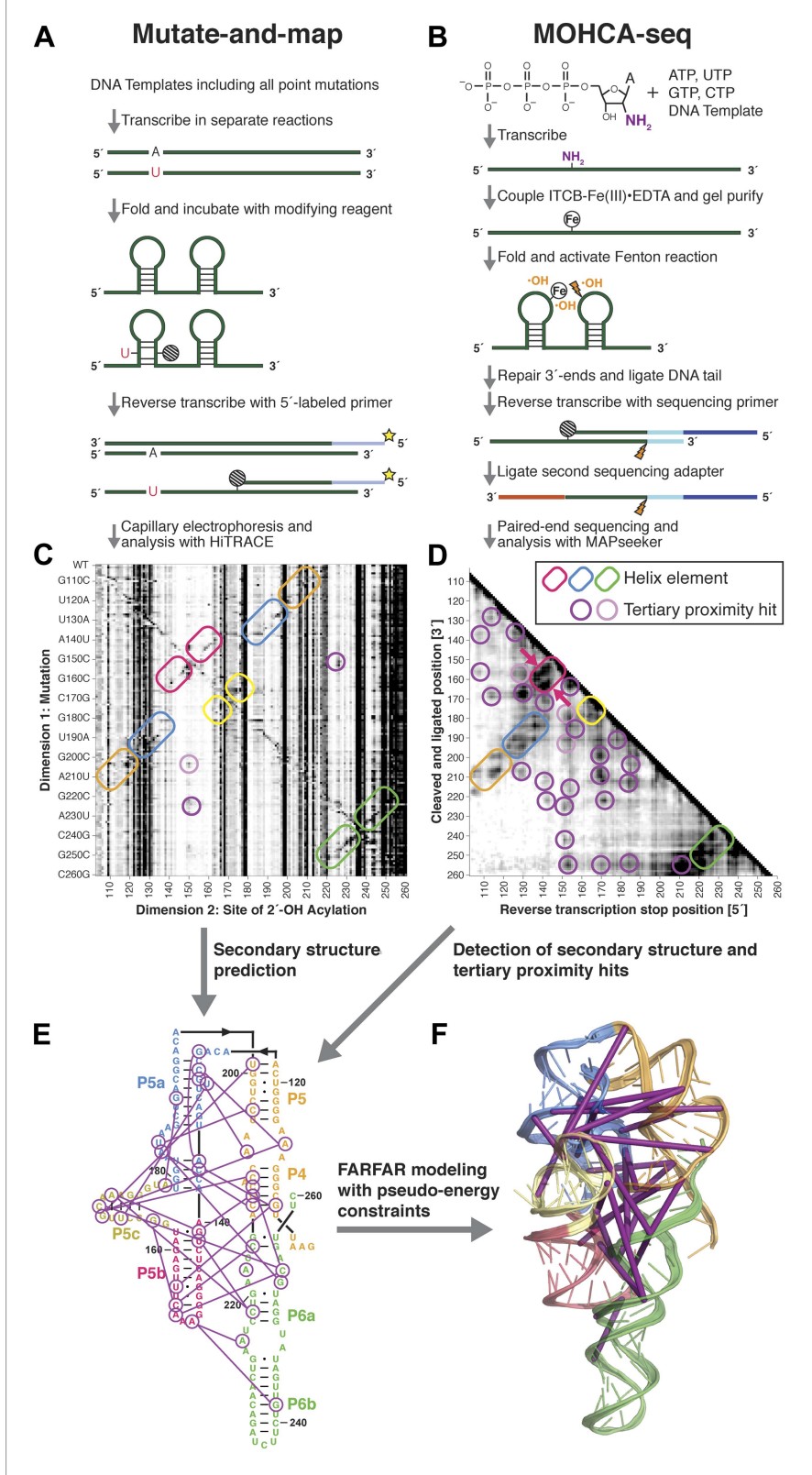

**Figure 1**. A multidimensional chemical mapping (MCM) pipeline to infer non-coding RNA 3D folds. (**A**) Schematic of mutate-and-map (M²) workflow, showing transcription of comprehensive point mutant library and chemical mapping of library with reverse transcription readout (*Kladwang et al., 2011a*). (**B**) Schematic of Multiplexed •OH (hydroxyl

*Figure 1. continued on next page*

*Figure 1. Continued*

radical) Cleavage Analysis (MOHCA)-seq workflow, showing random incorporation of radical sources, fragmentation, adapter ligations, and analysis by sequencing. (**C**) M$^2$ data set for P4–P6 domain of *Tetrahymena* group I ribozyme. (**D**) MOHCA-seq data set (proximity map) for P4–P6. In (**C**) and (**D**), rounded rectangles indicate helix elements with colors matching helices in (**E–F**), purple circles indicate hits corresponding to < 30 Å pairwise distance in the crystal structure, and pink circles indicate hits corresponding to > 30 Å pairwise distance in the crystal structure. In (**D**), magenta arrows indicate MOHCA-seq hits due to diffusion across the major (top arrow) or minor (bottom arrow) grooves from radical sources located in P5b. (**E–F**) Representation of MOHCA-seq tertiary proximities on M$^2$-guided secondary structure (**E**) and on final single Rosetta model (**F**). Purple lines indicate MOHCA-seq hits corresponding to < 30 Å pairwise distance in the crystal structure. *Figure 1—figure supplement 1* shows stages of MOHCA-seq data analysis in the MAPseeker software package (accessible through the RNA Mapping Database server at http://rmdb.stanford.edu/tools/). *Figure 1—figure supplement 2* shows the pseudo-energy potential used for pairwise MOHCA-seq constraints in Rosetta modeling and plots of MOHCA-seq signal vs pairwise distance.

The following figure supplements are available for figure 1:

**Figure supplement 1**. MOHCA-seq data analysis.

**Figure supplement 2**. Pseudo-energy potential used to incorporate MOHCA-seq constraints in Rosetta modeling, and plots of MOHCA-seq signal vs pairwise distance.

MOHCA-seq simplifies and accelerates the readout using techniques developed for RNA-seq (see, e.g., ref. [*Ingolia et al., 2012*]). To read out strand cleavage sites, the new workflow includes an end-repair step using T4 polynucleotide kinase under conditions that promote its 3′-phosphatase activity (*Cameron and Uhlenbeck, 1977*). This reaction permits ligation of a pre-adenylated universal adapter sequence to the 3′-ends of the hydroxyl radical-cleaved RNA fragments, allowing reverse transcription with primers harboring an Illumina adapter and experiment-specific barcodes (*Ingolia et al., 2012*). We expected reverse transcription to then stop at the nucleotide tethered to the bulky radical source at its C2′ position, providing a readout of the source position that is more convenient than the MOHCA-gel strand-scission method (*Figure 2A*, left). However, we were surprised to find additional stop sites beyond these source attachment positions (*Figure 2A*, right), suggesting that reverse transcription could also terminate at sites of additional oxidative damage events, including not only radical-induced strand scission but also other products of ribose or nucleobase oxidation (e.g., 8-oxo-purines and 5-hydroxy-pyrimidines) (*Yanagawa et al., 1990*, *1992*; *Rhee et al., 1995*; *Pogozelski and Tullius, 1998*; *Barciszewski et al., 1999*; *Gong et al., 2006*), which are indistinguishable at the cDNA level from stops at tethered radical sources. This high rate of stops would have been difficult to quantify in prior gel and capillary electrophoresis (CE) measurements (*Mitra et al., 2008*; *Kladwang et al., 2014*); the rate and proximity of these stops to the radical source were confirmed through both CE and next-generation sequencing tests on a model hairpin RNA with a single radical source attachment position (*Figure 2B–D*). The rate of reverse transcription-terminating oxidative damage events exceeds that of strand scission alone by many fold (*Figure 2E*). Since these sites of reverse transcription termination marked locations that were also nearby in three dimensions to the radical source, the 5′ and 3′ ends of the resulting cDNAs would still be expected to report pairs of sites that were proximal in the RNA structure, albeit at longer average distances than source-to-cleavage pairs (illustrated in *Figure 2A*). Thus, MOHCA-seq enables detection of correlated events previously invisible to MOHCA-gel, and we hypothesized that this significantly larger number of pairwise proximities would permit rapid ncRNA modeling with comparable accuracy to the prior method (*Figure 2A,F–G*).

After reverse transcription, the MOHCA-seq workflow is completed by ligating a second Illumina adapter to the cDNA library and then determining the 5′ and 3′ ends of each cDNA by paired-end sequencing on a standard Illumina platform (*Mortimer et al., 2012*; *Seetin et al., 2014*). MAPseeker software analysis quantifies the data and converts the raw counts into a pairwise proximity map ('Materials and methods' and *Figure 1—figure supplement 1*). Analogous to nuclear Overhauser spectroscopy for NMR structure determination, constraints selected from the analyzed data are then used to guide 3D computational modeling of the ncRNA using the Rosetta software (*Cheng et al., 2015*). Unlike MOHCA-gel, MOHCA-seq does not require specially synthesized nucleotides or 2D gel

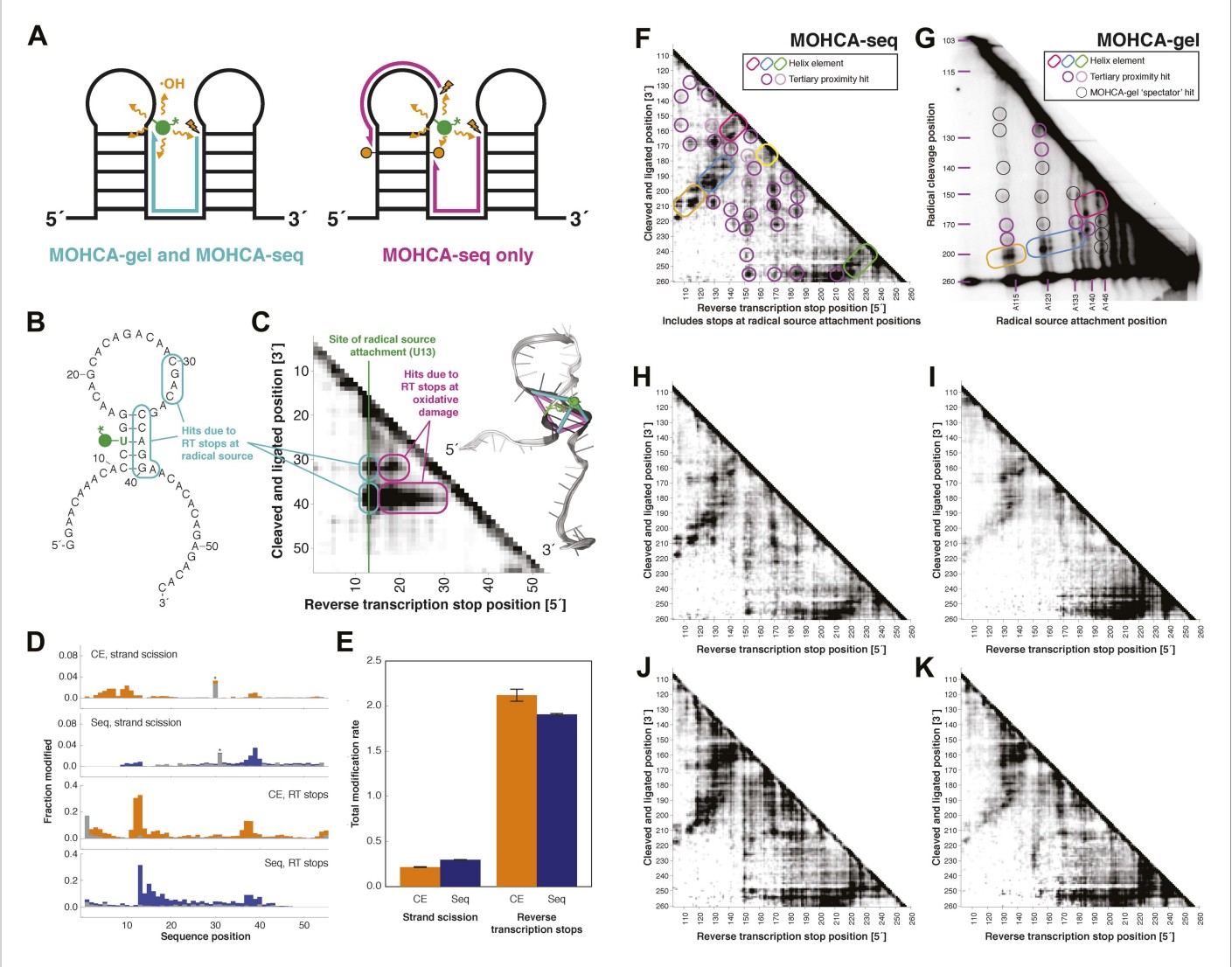

**Figure 2**. MOHCA-seq detects rich pairwise proximity information in single experiments and is robust to changes in the experimental protocol. (**A**) Schematic illustrating RNA fragments detectable by MOHCA-seq. At left, reverse transcription from a strand scission position (orange bolt) terminates at the radical source (in green); these fragments are also detectable by MOHCA-gel. At right, reverse transcription from strand scission positions can also terminate at additional oxidative damage events (orange circles) caused by the same radical source; these fragments are only detectable by MOHCA-seq. (**B**) Sequence and secondary structure of a proof-of-concept RNA with one radical source attachment site (U13, in green). Cyan rounded rectangles indicate expected MOHCA-seq hits due to RT stops at the radical source after cleavage at the circled residues. (**C**) MOHCA-seq proximity map showing expected hits (cyan rounded rectangles), as well as additional reverse transcription stops occurring 3′ of the radical source position but 5′ of the cleaved and ligated position, due to oxidative damage events 3′ of the radical source that are spatially correlated with the radical source (magenta rounded rectangles). At right, strand scission rates detected by paired-end sequencing (see (**D**)) are shown on a scale from white (low) to black (high) on a visualization-only model of the proof-of-concept RNA. Pairwise hits detectable by MOHCA-gel and MOHCA-seq (as in (**A**), left) are shown as cyan lines; hits detectable by MOHCA-seq only (as in (**A**), right) are shown as magenta lines. (**D**) Quantified strand scission and reverse transcription-terminating modification rates (note difference in scale) in the proof-of-concept RNA from capillary electrophoresis (CE) or paired-end sequencing data (Seq). Blue and orange bars are data for a MOHCA-seq sample with a tethered radical source, and gray bars are data for an identically treated control sample without a radical source. A '*' indicates a data point that was not included in calculation of the total rate of strand scission due to high background or bleed-through of a fluorescent reference ladder peak. (**E**) Total modification rates for strand scission or reverse transcription stops calculated from CE or sequencing data. (**F–G**) MOHCA-seq (**F**) and MOHCA-gel (**G**) proximity maps of P4–P6 from single experiments. MOHCA-gel data were collected previously according to the published method, with 5′-³²P-labeled P4–P6 RNA (***Das et al., 2008***). Rounded rectangles indicate helix elements, purple circles indicate hits corresponding to < 30 Å pairwise distance in the crystal structure, pink circles indicate hits corresponding to > 30 Å pairwise distance in the crystal structure, and black circles indicate MOHCA-gel 'spectator hits', which appear due to lack of rigorous background subtraction (***Das et al., 2008***; ***Kim et al., 2011***). The pink and purple circles in (**C**) correspond to the hits annotated in ***Figure 1D***, which shows the average of four data replicates; features that are not circled disappear after averaging. Note distortion of MOHCA-gel compared to MOHCA-seq due to nonlinearity of gel

*Figure 2. continued on next page*

**Figure 2. Continued**

electrophoresis rates with RNA length. (**H–K**) MOHCA-seq proximity maps for P4–P6 with alternative 2′-NH$_2$-modified nucleotides incorporated during transcription. Modified nucleotide triphosphate included in transcription reaction at molar ratio of 0.5 to unmodified NTP: (**H**) 2′-NH$_2$-2′-dATP; (**I**) 2′-NH$_2$-2′-dUTP; (**J**) 2′-NH$_2$-2′-dGTP; (**K**) 2′-NH$_2$-2′-dCTP. All fragmentation reactions were performed for 30 min. All four data sets were collected in one Illumina MiSeq run using a 50-cycle MiSeq Reagent Kit v2. Additional variations of the MOHCA-seq protocol are shown in *Figure 2—figure supplement 1* (variation of radical source incorporation rate) and *Figure 2—figure supplement 2* (variation of fragmentation reaction time).

The following figure supplements are available for figure 2:

**Figure supplement 1**. Variation of radical source incorporation rate.

**Figure supplement 2**. Variation of fragmentation reaction time.

electrophoresis; the high-throughput sequencing readout improves the signal-to-noise as well as the length windows visible per experiment; and the detection of previously invisible oxidation events enhances the mapping coverage of a single experiment. The protocol's throughput is also enhanced by its highly parallel nature, which we leveraged to co-load libraries for multiple RNAs on single sequencing runs ('Materials and methods').

## MOHCA-seq tertiary proximity map complements M$^2$ secondary structure for Rosetta modeling of P4–P6

We first tested the ability of MOHCA-seq to complete our M$^2$/Rosetta pipeline by applying it to a well-studied model system with known crystallographic structure, the 158-nucleotide P4–P6 domain of the *Tetrahymena* ribozyme (*Cate et al., 1996*) (sequence listed in *Supplementary file 1*). In comparison to prior MOHCA-gel data in which a single experiment yielded a few proximity pairs, raw sequencing counts from a single MOHCA-seq experiment exhibited dozens of pairwise proximity 'hits' throughout the ncRNA (*Figure 2F–G*). Additionally, the digital form of MOHCA-seq data enabled computational analysis to resolve further features. Our analysis, called closure-based •OH correlation analysis (COHCOA), is described fully in the 'Materials and methods'. Briefly, COHCOA assumes a two-dimensional background in the raw aligned sequencing data, which arises from two one-dimensional background profiles that are uncorrelated with tethered radical sources: a strand scission profile (leading to horizontal striations) and a reverse transcription-terminating profile (leading to vertical striations) (*Figure 1—figure supplement 1A*). Through iterative fitting, COHCOA calculates this two-dimensional background and subtracts it from the raw data and additionally corrects for reverse transcription attenuation. The procedure isolates the MOHCA-seq signal, which arises from oxidative damage that is spatially correlated with tethered radical sources (*Figure 1—figure supplement 1B*). To prevent false positive identification of MOHCA-seq hits due to noise in areas of the proximity map with fewer reads and therefore less signal, we apply a two-dimensional smoothing algorithm to produce the final proximity map for visualization (*Figure 1—figure supplement 1C*).

A comparison of a single COHCOA-analyzed MOHCA-seq data set to a single MOHCA-gel experiment is shown in *Figure 2F–G*, and a final MOHCA-seq data set consisting of four averaged replicates is shown in *Figure 1D*. The P4–P6 RNA has five major helical secondary structure elements, P4, P5a, P5b, P5c, and P6, which were originally characterized at high resolution by crystallography and recently confirmed to occur in solution at base-pair resolution by M$^2$ experiments (*Kladwang et al., 2011a*). Indeed, MOHCA-seq proximity hits corresponded to radicals diffusing across the major and minor grooves of each of these helices (visible as two separate branches, arrows in *Figure 1D*), confirming the M$^2$ secondary structure, albeit at lower resolution (compare *Figure 1C,D*). Importantly, nearly two dozen tertiary proximities, including features marking the bend at the J5/5a hinge, the A-minor contact between the A-rich bulge and P4, and the docking of the L5b tetraloop into the J6a/6b receptor, appeared as punctate hits (*Figure 1D–F*); only the last of these was clearly observed in M$^2$ mapping (*Figure 1C*).

To test the robustness of the protocol, we varied several experimental parameters: (1) the 2′-NH$_2$-modified nucleotide incorporated during transcription, (2) the radical source incorporation rate, and (3) the fragmentation reaction time. We observed MOHCA-seq features at similar positions (but with

variable intensities) whether radical sources were attached at A, C, G, or U (*Figure 2H–K*); whether the ratio of 2′-NH$_2$-modified nucleotides to standard nucleotides was 1:5, 1:2, or 5:4 (*Figure 2—figure supplement 1*); or whether the fragmentation reaction time was 5, 10, or 30 min (*Figure 2—figure supplement 2*). In particular, even after 30 min of radical source activation, the proximity information in the experimental data remains clearly visible without noticeable increase in background, suggesting that the radical source cleaves itself off the RNA in minutes, limiting the reaction. CE of ascorbate-treated end-labeled RNA supported this mechanism of self-limiting the reaction (see also *Das et al., 2008*). A few weak signals in the proximity maps that were not predicted by the crystal structure (pink circles, *Figure 1D*) may be caused by perturbations of the ncRNA structure by radical source attachment or by conformational dynamics of the RNA or the ~11 Å benzyl-thiourea linker tethering the radical source to the RNA backbone.

Based on previous MOHCA-gel studies, we expected that our MOHCA-seq data would enable 3D computational modeling of the P4–P6 domain to helical resolution or better without crystallographic information. We used the FARNA with full-atom refinement algorithm in Rosetta (*Das et al., 2010*) for modeling, described in detail in the 'Materials and methods'. First, to reduce the computational expense of sampling helices, which primarily adopt A-form conformations, we pre-assembled secondary structure elements in Rosetta. To mimic the lack of high-resolution data for cases of blind RNA structure prediction, we used helices derived from M$^2$ and 1D SHAPE mapping for modeling. Second, we performed global modeling in Rosetta in two stages. We generated an initial set of over 10,000 low-resolution models using FARNA in parallel, independent modeling runs; then, we minimized the 1/6 of the models with the lowest Rosetta energy score using full-atom refinement. In both stages, we included MOHCA-seq data to constrain sampling.

To apply the MOHCA-seq data in Rosetta, we generated a list of pairs of residues that were suggested to be proximal by the data by identifying peaks that were (1) distinguishable from the local background signal and (2) not attributable to secondary structure, based on secondary structures inferred by M$^2$ (*Kladwang et al., 2011a*) (purple and pink circles, *Figure 1D* and tabulated in *Supplementary file 2*). As in prior work (*Das et al., 2008*), we assumed that signals due to dynamic heterogeneity would be weak compared to those due to stable tertiary proximities. Therefore, we used all data, including the few weak features not explained by the crystal structure (pink circles, *Figure 1D*), anticipating that the modeling would automatically determine the maximal subset of contacts consistent with each other and with a single structure. Conformational sampling in Rosetta was constrained by applying a smooth pseudo-energy potential (*Figure 1—figure supplement 2A*) between each pair of residues in the list that favored conformations with the residues between 0 and 30 Å apart and disfavored conformations with the residues more than 30 Å apart. This potential, which was similar to that developed previously for MOHCA-gel analysis, was suggested based on observed correlation of the MOHCA-seq intensity to distances in residue pairs involved in secondary structures (*Figure 1—figure supplement 2B* and ref. [*Das et al., 2008*]) and was validated below.

Finally, we selected a single representative model by clustering the lowest-energy 3% of the refined models using Rosetta. An all-heavy-atom RMSD threshold was chosen to give 1/6 of the clustered models in the most populated cluster, as in prior work (*Das et al., 2008*). As in protein structure modeling (*Shortle et al., 1998*), the final single model of this MCM pipeline is the model with the largest number of neighboring models within the RMSD threshold, called the 'cluster center', which is not necessarily the model with the lowest Rosetta energy score. The cluster center represents the best tertiary structure prediction of the MCM pipeline for automated Rosetta modeling guided by experimental data. The RMSD threshold used for clustering represents the intra-cluster RMSD between models in the cluster and is used to calculate an in situ RMSD estimate, representing the precision of modeling (see 'Materials and methods'). The accuracy of the MCM model was calculated as the all-heavy-atom RMSD between the cluster center and the crystal structure, as in our previous study (*Das et al., 2008*).

For the P4–P6 benchmark case, we found that the cluster center of the MCM pipeline had an accuracy of 8.6 Å RMSD to the crystal structure (*Figures 1F*, *3A* and *Table 1*), which was within the 16.5 Å precision of the modeling estimated from the clustering threshold (*Table 2*). This suggests that MCM is capable of defining the global 3D structure of the RNA, without crystallographic information, at near the resolution of a single-nucleotide register shift (6 Å) even when weak MOHCA-seq features not explained by the crystal structure are used to constrain modeling. Performing the same Rosetta modeling protocol without MOHCA-seq pairwise constraints led to a strikingly worse model with

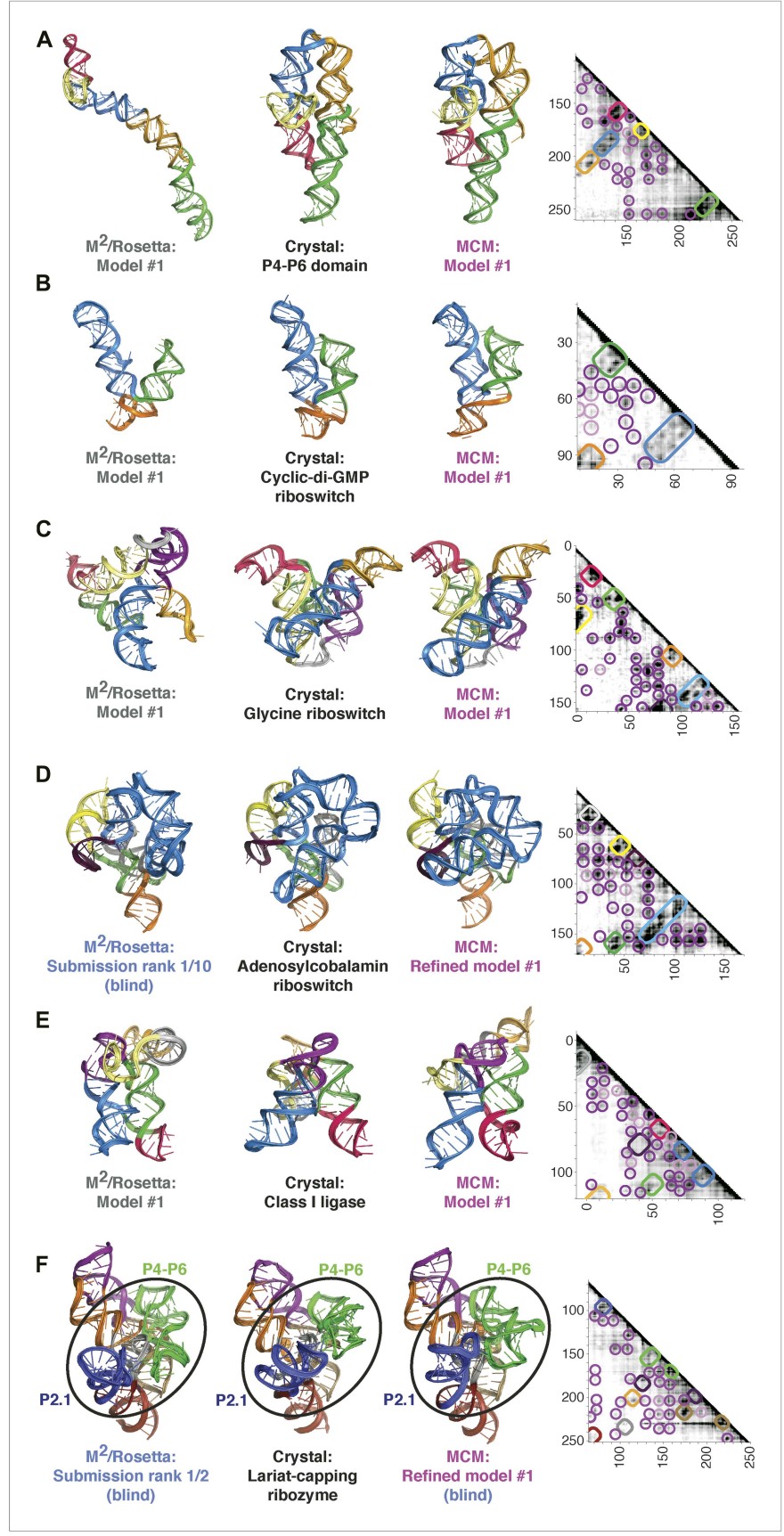

Figure 3. continued on next page

*Figure 3. Continued*

**Figure 3**. MCM achieves 1-nm resolution models of complex RNA folds. (**A**) P4–P6 domain: M$^2$/Rosetta model (left), 38.3 Å root-mean-squared-deviation (RMSD); crystal structure (PDB ID 1GID, center left); M$^2$/MOHCA-seq/Rosetta (MCM) model (center right), 8.6 Å RMSD. (**B**) *V. cholerae* cyclic-di-GMP riboswitch aptamer: M$^2$/Rosetta model (left), 11.3 Å RMSD; crystal structure (PDB ID 3IRW, center left); MCM model (center right), 7.6 Å RMSD. (**C**) *F. nucleatum* double glycine riboswitch ligand-binding domain: M$^2$/Rosetta model (left), 30.5 Å RMSD; crystal structure (PDB ID 3P49, center left); MCM model (center right), 7.9 Å RMSD. (**D**) *S. thermophilum* adenosylcobalamin (AdoCbl) riboswitch aptamer: M$^2$/Rosetta submission rank 1 of 10 for RNA-puzzle 6 (left), 17.1 Å RMSD; crystal structure (PDB ID 4GXY, center left); MCM model (center right), 11.9 Å RMSD. (**E**) Class I ligase: M$^2$/Rosetta model (left), 26.3 Å global RMSD, and 14.0 Å core RMSD; crystal structure (PDB ID 3HHN, center left); MCM model (center right), 14.5 Å global RMSD and 11.1 Å core RMSD. (**F**) *D. iridis* lariat-capping ribozyme: M$^2$/Rosetta submission rank 1 of 2 for RNA-puzzle 5 (left), 17.0 Å P2.1/P4–P6 RMSD and 9.6 Å global RMSD; crystal structure (PDB ID 4P8Z, center left); MCM model (center right), 11.2 Å P2.1/P4–P6 RMSD and 8.2 Å global RMSD. In (**A–F**), MOHCA-seq proximity maps with annotated helix elements (rounded rectangles) and tertiary hits (purple and pink circles) as in *Figure 1D* are shown at right. Full-size proximity maps, including 5′- and 3′-flanking sequences outside the region of interest, are shown in *Figure 3—figure supplement 3*. M$^2$ analyses of AdoCbl riboswitch aptamer, class I ligase, and lariat-capping ribozyme are shown in *Figure 3—figure supplements 1, 2, 7*. *Figure 3—figure supplement 4* shows comparisons of models generated by different computational methods for RNA-puzzle 6. *Figure 3—figure supplements 5, 6* show comparisons of additional modeling runs for class I ligase and AdoCbl riboswitch aptamer to crystal structures.

The following figure supplements are available for figure 3:

**Figure supplement 1**. M$^2$ analysis of the *S. thermophilum* adenosylcobalamin riboswitch aptamer, performed during the sixth RNA-puzzles structure prediction trial.

**Figure supplement 2**. M$^2$ analysis of class I ligase.

**Figure supplement 3**. Full MOHCA-seq proximity maps, including 5′- and 3′-flanking regions.

**Figure supplement 4**. Blind models generated for RNA-puzzles can attain 1-nm resolution or better but cannot predict the most accurate models.

**Figure supplement 5**. Comparison of class I ligase crystal structure and knotted and unknotted MCM models.

**Figure supplement 6**. Comparison between MCM models of the adenosylcobalamin riboswitch aptamer using different initial RNA fragment sets.

**Figure supplement 7**. M$^2$ analysis of the GIR1 lariat-capping ribozyme, performed during the fifth RNA-puzzles structure prediction trial.

38.3 Å accuracy to the crystal structure (*Figure 3A* and *Table 1*), demonstrating that MOHCA-seq constraints are required for accurate modeling. The 8.6 Å accuracy of the cluster center was better than the previous 13 Å accuracy of modeling guided by MOHCA-gel data, which were collected at substantially greater experimental expense and did not allow base-pair resolution inference of secondary structure, as is now provided by M$^2$ analysis (*Das et al., 2008*).

The cluster center MCM model correctly inferred the global 3D structure of the P4–P6 domain, including the compaction of the P5c helix and A-rich bulge toward P4 and the clothespin-like bend allowing proximity of the L5b tetraloop and the J6a/6b receptor. Nevertheless, we found that the resolution was not sufficient to recapitulate nucleotide-level details, such as the base stacking and base–triple interactions in the tetraloop/tetraloop receptor contact. If the P4–P6 domain were a blind prediction case, the MOHCA-seq data and MCM model would have marked these regions as sites for further biochemical investigation, but accurate recovery of the detailed interactions would still require advances in computational methods being developed for refining conformations of motifs to high resolution (*Sripakdeevong et al., 2011*). Additionally, not all MOHCA-seq constraints were satisfied by either our cluster center MCM model or the crystal structure, an observation discussed further below.

**Table 1.** Benchmark of MCM on RNAs with crystal structures

| RNA | Length | M²/Rosetta (no MOHCA, control) | | MCM | |
| --- | --- | --- | --- | --- | --- |
| | | RMSD to crystal (Å) (accuracy) | p-value§ | RMSD to crystal (Å) (accuracy) | p-value§ |
| *Tetrahymena* ribozyme P4–P6 domain | 158 | 38.3 | >0.9 | 8.6 | <1.0 × 10⁻¹⁶ |
| V. cholerae cyclic-di-GMP riboswitch aptamer, ligand-bound* | 89 | 11.3 | 2.6 × 10⁻³ | 7.6 | 6.3 × 10⁻⁷ |
| F. nucleatum double glycine riboswitch ligand-binding domain, ligand-bound* | 159 | 30.5 | >0.9 | 7.9 | <1.0 × 10⁻¹⁶ |
| S. thermophilum adenosylcobalamin riboswitch aptamer, ligand-bound* | 168 | 17.1† | 5.3 × 10⁻⁷ | 11.9 | 4.0 × 10⁻¹⁵ |
| Class I ligase | 127 | 26.3 | >0.9 | 14.5 | 6.8 × 10⁻⁵ |
| Class I ligase, core domain‡ | 87 | 14.0 | 0.13 | 11.1 | 3.1 × 10⁻³ |
| D. iridis lariat-capping ribozyme | 188 | 9.6† | <1.0 × 10⁻¹⁶ | 8.2 | <1.0 × 10⁻¹⁶ |
| *D. iridis* lariat-capping ribozyme, MCM refined regions‡ | 69 | 17.0† | n.a.§ | 11.2 | n.a.§ |

*MCM modeling was performed with MOHCA-seq constraints from datasets collected on the ligand-bound state; ligands were not included during Rosetta modeling.

†M²/Rosetta statistics are reported for RNA-puzzle submission rank 1 models, which included subdomains built by homology modeling.

‡Calculated over core domain residues or refined regions after alignment using MAMMOTH (*Ortiz et al., 2002*); see 'Materials and methods'.

§p-value computed using analytical formula for secondary-structure-constrained 3D modeling in (*Hajdin et al., 2010*); it is not applicable to peripheral domains. Value above 0.9 are not well-determined and are presented as > 0.9.

MCM: multidimensional chemical mapping; RMSD: root-mean-squared-deviation.

## MCM allows rapid modeling of diverse ncRNAs, including blind prediction challenges

Our initial studies on the P4–P6 RNA suggested that the MCM pipeline, consisting of M² and MOHCA-seq experiments followed by Rosetta modeling, could be a generally applicable approach for determining global structures of ncRNAs. To further test this hypothesis, we benchmarked MCM on several additional ncRNAs with known structures but distinct functions: the 89-nucleotide *Vibrio cholerae* cyclic-di-GMP (c-di-GMP) riboswitch aptamer bound to its ligand at 10 µM c-di-GMP (*Smith et al., 2009*), the 159-nucleotide ligand-binding domain of the *Fusobacterium nucleatum* double glycine riboswitch (consisting of two tandem glycine aptamers) bound to two glycines at 10 mM glycine (*Butler et al., 2011*), the 168-nucleotide *Symbiobacterium thermophilum* adenosylcobalamin (AdoCbl) riboswitch aptamer bound to its ligand at 70 µM AdoCbl (*Peselis & Serganov, 2012*), and the 127-nucleotide class I ligase including substrate obtained from in vitro selection (*Ekland and Bartel, 1995*; *Shechner et al., 2009*) (sequences listed in *Supplementary file 1*). For the glycine riboswitch ligand-binding domain, the AdoCbl riboswitch aptamer, and the class I ligase, the secondary structures were first defined at nucleotide resolution by high-throughput M² analysis; data from this previously established approach were collected in prior work (*Kladwang et al., 2011a*) or are shown in *Figure 3—figure supplements 1, 2*. MOHCA-seq data and Rosetta modeling completed the MCM pipeline (*Figure 3B–E* and *Figure 3—figure supplement 3B–E*). As described above for the P4–P6 RNA and in prior work on 3D ncRNA modeling, precision estimates were possible in situ without knowledge of the actual structure by assessing convergence of independent modeling runs ('Materials and methods'); all cases suggested 1-nm resolution had been achieved (*Table 2*). Indeed, this resolution was attained by MCM in all cases: 7.6 Å for the c-di-GMP riboswitch aptamer; 7.9 Å for the glycine riboswitch ligand-binding domain; 11.9 Å for the AdoCbl riboswitch aptamer (compare to RNA-puzzle 6 models, *Figure 3—figure supplement 4*); and 11.1 Å (ribozyme) and 14.5 Å (ribozyme including substrate) for the class I ligase (*Shechner et al., 2009*) (*Figure 3B–E* and *Table 1*; clusters with knots were not accepted, see *Figure 3—figure supplement 5* and *Table 2*). The lack of correlation between the fraction of satisfied MOHCA-seq pairwise constraints and the accuracy of the models

**Table 2.** Precision and constraint statistics of MCM modeling, including conformationally heterogeneous states

| RNA | Length | Number of models | Size of largest cluster | RMSD to crystal, cluster center (Å) (accuracy) | In situ RMSD estimate (Å) (precision) | Total constraints | Percent strong constraints satisfied,* cluster center | Percent strong constraints satisfied, crystal | Percent weak constraints satisfied, cluster center | Percent weak constraints satisfied, crystal |
|---|---|---|---|---|---|---|---|---|---|---|
| Tetrahymena ribozyme P4–P6 domain | 158 | 61,115 | 48 | 8.6 | 16.5 | 35 | 65.4 | 69.2 | 55.6 | 44.4 |
| V. cholerae cyclic-di-GMP riboswitch aptamer, bound | 89 | 15,632 | 10 | 7.6 | 6.8 | 19 | 100.0 | 100.0 | 100.0 | 75.0 |
| V. cholerae cyclic-di-GMP riboswitch aptamer, unbound† | 89 | 26,421 | 5 | (26.0)‡ | 9.8 | 14 | 100.0 | 83.3 | 75.0 | 37.5 |
| F. nucleatum double glycine riboswitch ligand-binding domain, bound | 159 | 16,506 | 14 | 7.9 | 10.0 | 31 | 95.8 | 100.0 | 57.1 | 42.9 |
| F. nucleatum double glycine riboswitch ligand-binding domain, unbound† | 159 | 15,771 | 12 | (25.4)‡ | 26.7 | 12 | N/A§ | N/A§ | 58.3 | 83.3 |
| F. nucleatum double glycine riboswitch ligand-binding domain with leader, bound | 167 | 23,153 | 19 | 11.8 | 10.6 | 49 | 89.5 | 84.2 | 83.3 | 83.3 |
| F. nucleatum double glycine riboswitch ligand-binding domain with leader, unbound† | 167 | 18,096 | 15 | (15.6)‡ | 14.7 | 34 | 77.8 | 33.3 | 64.0 | 68.0 |
| S. thermophilum adenosylcobalamin riboswitch aptamer, bound# | 168 | 14,219 | 12 | 11.9 | 13.9 | 38 | 84.0 | 76.0 | 84.6 | 69.2 |
| S. thermophilum adenosylcobalamin riboswitch aptamer, unbound†,# | 168 | 11,980 | 10 | (17.3)‡ | 19.7 | 33 | 83.3 | 94.4 | 73.3 | 73.3 |
| Class I ligase (unknotted) | 127 | 17,881 | 7 | 14.5 | 11.6 | 24 | 86.7 | 53.3 | 22.2 | 77.8 |
| Class I ligase (unknotted), core domain¶ | 87 | 17,881 | 7 | 11.1 | 12.0 | 18 | 91.7 | 58.3 | 33.3 | 50.0 |

*Table 2. Continued on next page*

*Table 2. Continued*

| RNA | Length | Number of models | Size of largest cluster | RMSD to crystal, cluster center (Å) (accuracy) | In situ RMSD estimate (Å) (precision) | Total constraints | Percent strong constraints satisfied,* cluster center | Percent strong constraints satisfied, crystal | Percent weak constraints satisfied, cluster center | Percent weak constraints satisfied, crystal |
|---|---|---|---|---|---|---|---|---|---|---|
| Class I ligase (knotted) | 127 | 17,881 | 14 | 16.1 | 11.6 | 24 | 86.7 | 53.3 | 55.6 | 77.8 |
| D. iridis lariat capping ribozyme | 188 | 19,741 | 16 | 8.2 | 5.7 | 32 | 90.0 | 70.0 | 54.5 | 54.5 |
| D. iridis lariat capping ribozyme, RNA-puzzles submission model #1 (2012) | 188 | – | – | 9.6 | – | – | 60.0 | 70.0 | 59.1 | 54.5 |
| D. iridis lariat capping ribozyme, MCM refined regions¶ | 69 | 19,741 | 16 | 11.2 | 10.5 | 32 | 87.5 | 62.5 | 46.2 | 38.5 |
| D. iridis lariat capping ribozyme, RNA-puzzles submission model #1 (2012), MCM refined regions# | 69 | – | – | 17.0 | – | – | 50.0 | 62.5 | 46.2 | 38.5 |
| HoxA9 5′-UTR 957–1132 domain, pseudoknot | 176 | 15,580 | 12 | – | 19.5 | 20 | 41.7 | – | 50.0 | – |
| HoxA9 5′-UTR 957–1132 domain, no pseudoknot | 176 | 11,234 | 8 | – | 19.8 | 21 | 33.3 | – | 22.2 | – |

*Constraints were considered satisfied if the O2′ of the 5′-residue was less than 30 Å from the C4′ of the 3′-residue (**Supplementary file 2**).

†For ligand-free states of the adenosylcobalamin riboswitch aptamer, cyclic-di-GMP riboswitch aptamer, and glycine riboswitch ligand-binding domain, RMSDs and percent constraints satisfied were calculated for gold-standard crystal structures solved in the presence of ligand.

‡For states not expected to agree with crystal structures, RMSD accuracy is given in parentheses.

§No strong constraints were selected for the unbound state of the glycine riboswitch ligand-binding domain without leader sequence.

#Three separate modeling runs were performed for each ligand-binding state of the riboswitch aptamer, using distinct sets of prebuilt fragments prepared for RNA-puzzle 6; see 'Materials and methods'. The number of models, size of largest cluster, cluster center RMSD (accuracy), and constraint satisfaction percentages are representative data from the modeling run with the most models generated. The intra-cluster RMSD is calculated as the mean pairwise RMSD between the cluster centers of the top cluster from each of the three modeling runs.

¶Calculated over refined regions or core domain residues after alignment using MAMMOTH; (**Ortiz et al., 2002**) see 'Materials and methods'.

MCM: multidimensional chemical mapping; RMSD: root-mean-squared-deviation.

(*Table 2*) suggests that unsatisfied constraints are due to conformational fluctuations in the RNA or radical-source linkers around the dominant global structures of these molecules, rather than systematic structural shifts or incomplete computational sampling. The generally consistent accuracy of these predictions supports our hypothesis that highly sampled computational modeling is able to filter out such inconsistent constraints, as has been observed in prior macromolecule modeling based on pairwise information (*Bowers et al., 2000*; *Das et al., 2008*; *Thompson et al., 2012*). Further supporting the assumptions of modeling, including the form of the pseudo-energy function, the relationship of MOHCA-seq signal intensities to pairwise distances seen in crystal structures was recovered by MCM models (*Figure 1—figure supplement 2B,C*). Again, we tested the necessity of MOHCA-seq data for completing the MCM pipeline by performing identical modeling of all ncRNA domains without MOHCA-seq constraints ($M^2$/Rosetta modeling). Just as for the P4–P6 domain, these runs produced dramatically less accurate models, as can be visually assessed (*Figure 3*) and quantitated by both RMSD to the crystal structure and statistical significance (*Hajdin et al., 2010*) (*Table 1*).

In addition to generating accurate models, the full MCM pipeline enabled selection of *single* confident 1-nm resolution tertiary structures in each of the above cases, an achievement highlighted by the AdoCbl riboswitch aptamer test case. For this RNA-puzzle target, we previously collected and disseminated $M^2$ data (*Figure 3—figure supplement 1*) that correctly discriminated between literature models of this ncRNA's secondary structure (*Peselis & Serganov, 2012*). However, these data did not disambiguate between a range of 3D structures, and our ten submitted models gave RMSDs from 12.1 Å to 32.8 Å to the subsequently released riboswitch aptamer crystal structure (*Figure 3—figure supplement 3*). With the developments in this study, we suspected that MOHCA-seq data would complement our $M^2$/Rosetta method and permit confident selection and submission of a single model. Indeed, the full MCM pipeline resolved tertiary contacts between the peripheral and core regions of the ncRNA (*Figure 3D*), and automated modeling converged to a structure defined with 13.9 Å resolution (in situ precision estimate), which achieved 11.9 Å RMSD to the crystal structure (*Figure 3D*, *Table 1*, *Figure 3—figure supplement 6*, and 'Materials and methods'), comparable to the accuracy of our best RNA-puzzle submission and clearly defining the global structure to helix resolution.

As a rigorous test, we applied MCM to model another RNA-puzzle before the release of its crystal structure, a 188-nucleotide lariat-capping ribozyme from *Didymium iridis*. As part of this blind challenge, we previously collected and shared $M^2$ data (*Figure 3—figure supplement 7*) that supported literature models of the ncRNA's secondary structure and suggested two additional tertiary contacts creating a 'ring' around the ribozyme. Nevertheless, during the RNA-puzzle time window in the summer of 2012, modeling from $M^2$ alone was uncertain, particularly in the long, kissing helical stacks P2.1 and P4–P6, which had no homologies to any experimentally solved structure (blue and green regions, *Figure 3F*). After developing MOHCA-seq but before the release of the lariat-capping ribozyme crystal structure, we completed the MCM pipeline by acquiring MOHCA-seq data for the ncRNA (*Figure 3F*), and the resulting map highlighted potential errors in these extended peripheral regions. We then blindly refined the peripheral regions based on the full MCM data (*Figure 3F* and *Table 1*), and the subsequent release of the ribozyme crystal structure (*Meyer et al., 2014*) confirmed that their accuracy significantly improved, from an original RMSD of 17.0 Å (original RNA-puzzle submission, $M^2$/Rosetta) to a final RMSD of 11.2 Å (full MCM, i.e., $M^2$/MOHCA-seq/Rosetta). The global accuracy of the models also improved, with a range of RMSDs over the full ribozyme from 7.6 to 8.9 Å, compared to our original submission, which turned out to have an overall RMSD accuracy of 9.6 Å. Taken together, these results rigorously demonstrated the ability of the full MCM pipeline to detect errors and to consistently and blindly refine ncRNA structures to 1-nm resolution or better throughout complex structures.

## Direct nucleotide-resolution observation of preformed riboswitch aptamer tertiary structure using MCM

Riboswitches form a major class of ncRNAs, and understanding their structures in the absence of ligands is critical for dissecting mechanisms of ligand capture and modulation of gene expression. While crystal structures and solution scattering of a few ligand-free riboswitch aptamers have suggested the possibility of extensive preformed tertiary structure in special cases (*Serganov et al., 2008*; *Baird et al., 2010*; *Wood et al., 2012*), whether these structures are generally sampled in

solution in the absence of ligand has remained poorly understood due to the lack of an unbiased, high-throughput solution structural technique. We tested if MCM could resolve previously missing information on ligand-free states of the three riboswitch aptamers that were characterized above in their ligand-bound forms. Because the computational modeling of heterogeneous conformational ensembles from experimental data is an open problem (*Beauchamp et al., 2014*; *Salmon et al., 2014*; *Shi et al., 2014*), our primary conclusions below are based on the M$^2$ and MOHCA-seq data themselves, with Rosetta modeling carried out for initial visualization only (see 'Materials and methods').

We first applied MCM to the c-di-GMP riboswitch aptamer without its ligand, which had been previously studied with state-of-the-art small-angle X-ray scattering, single-molecule FRET, and kinetic analysis (*Kulshina et al., 2009*; *Wood et al., 2012*). These prior studies revealed an extended conformation of this ncRNA transiently interconverting with a 'docked' conformation that shares at least one tertiary contact with the ligand-bound docked state (P1b/P2 tetraloop/receptor). MCM rapidly recovered this picture and provided additional information. In terms of secondary structure, M$^2$ measurements gave P1a, P1b, and P2 helices in both ligand-free and ligand-bound states, up to possible rearrangements in edge base pairs (*Kladwang et al., 2011a*), and MOHCA-seq additionally confirmed all 3 helices in both states (*Figure 4A*). In terms of tertiary structure, unambiguous proximities between several segments of P1b and P2 that were detected in the ligand-bound state (*Figure 4A*, left) were also observed at the same locations, albeit weakly, in the ligand-free state (*Figure 4A*, right). To test that these tertiary features were not due to noise or experimental artifacts in MOHCA-seq experiments or data processing, we confirmed that they disappeared in solution conditions without Mg$^{2+}$ (*Figure 4B*). The matching 'fingerprints' of cross-helix tertiary features in ligand-free and ligand-bound MOHCA-seq maps provided evidence that the ligand-free riboswitch aptamer samples the same tertiary conformation as the ligand-bound docked state at nanometer resolution, including and going beyond the known tetraloop/receptor contact.

To visualize the ensemble, we performed Rosetta modeling as above and discovered that the conformational heterogeneity of the top cluster was significantly larger than clustering of ligand-bound states above (9.8 Å vs 6.8 Å, respectively; *Table 2*), suggesting that a significant fraction of the visible variance (*Figure 4C*) arises from actual RNA conformational dynamics rather than from modeling uncertainties. As in NMR studies of dynamic molecules, we therefore inspected an ensemble of models with the lowest Rosetta energy score instead of focusing on the cluster center; the resulting ensemble was used for initial visualization only (*Al-Hashimi, 2007*; *Bothe et al., 2011*; *Salmon et al., 2014*). Four of the five lowest energy models were primarily extended and without tertiary contacts, with one showing a more compact state similar to the one stabilized by ligand. The MOHCA-seq data are thus consistent with a picture of the ligand-free state of the c-di-GMP riboswitch aptamer primarily adopting extended conformations but also interconverting to more compact ones (*Figure 4C*).

MCM analysis of riboswitch aptamers for glycine and adenosylcobalamin gave surprisingly similar results to our investigation of the c-di-GMP riboswitch aptamer. The glycine riboswitch ligand-binding domain consists of two glycine-binding aptamers connected by a linker; the formation of cross-aptamer contacts upon ligand binding could explain the cooperativity of glycine binding between the aptamers (*Erion and Strobel, 2011*; *Kladwang et al., 2012a*; *Esquiaqui et al., 2014*; *Ruff and Strobel, 2014*), but has not been directly visualized. In terms of secondary structure, previous M$^2$ data indicated that three major helical elements P1, P2, and P3 were formed by each of the two aptamers, both with and without glycine (*Kladwang et al., 2011a*); MOHCA-seq data gave additional support for this six-helix secondary structure being preformed without ligand (*Figure 5A*). Strikingly, tertiary proximity hits observed both within and across the two aptamers in the glycine-bound state were retained, though weakened, in the absence of glycine (*Figure 5A* and *Figure 3—figure supplement 3H*). Analogous to our control experiments for the c-di-GMP riboswitch aptamer, these tertiary proximities were not detectable in conditions without Mg$^{2+}$ (*Figure 5B*). Completing the MCM pipeline, Rosetta modeling of the ligand-free state with M$^2$/MOHCA-seq constraints gave a structural ensemble with an approximately similar global structure but substantially more conformational variance than the ligand-bound state (*Figure 5C* and *Table 2*). These MCM models provide a framework for further dissection of the ligand-free state of the glycine riboswitch aptamers through mutation and MCM. As an example, we discovered that the preformed tertiary proximities detected by MCM in the ligand-free riboswitch aptamer depend critically on at least one junction element, a leader sequence that was missed in

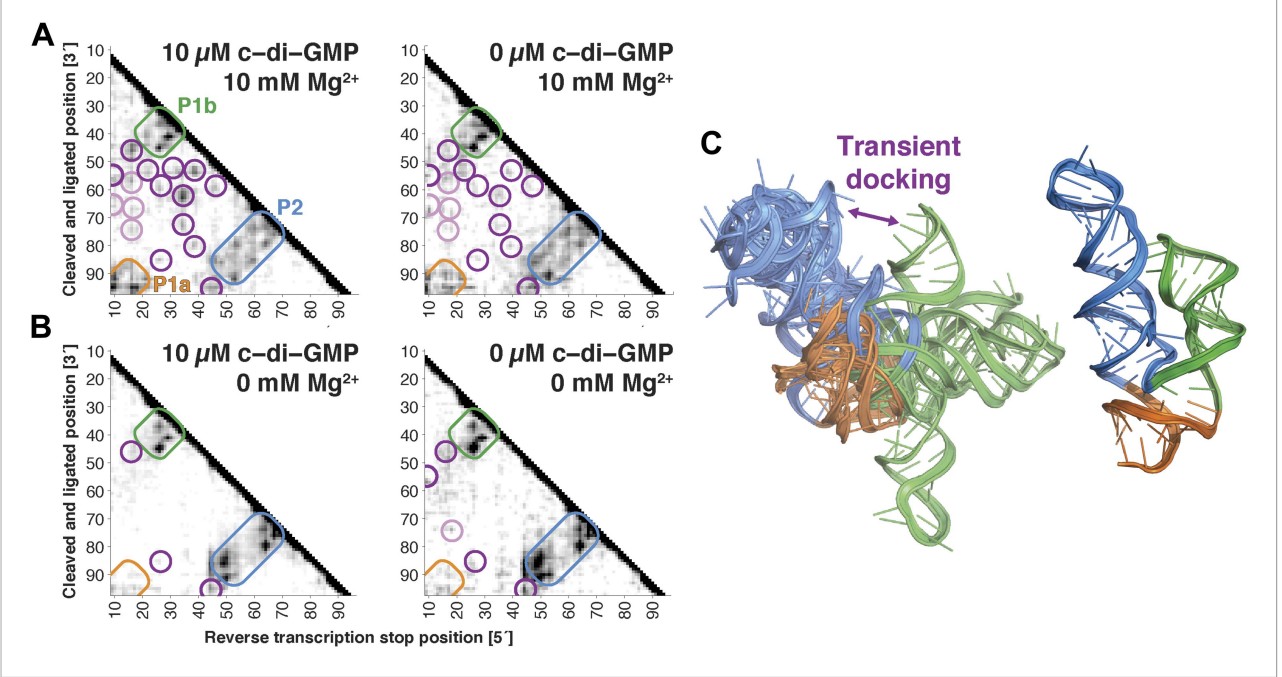

**Figure 4**. The *V. cholerae* cyclic-di-GMP riboswitch aptamer exhibits Mg²⁺-dependent preformed tertiary structure in its ligand-free state. (**A**) Proximity maps of c-di-GMP riboswitch aptamer in 10 μM (left) or 0 μM (right) c-di-GMP, folded in 10 mM Mg²⁺, with annotated helix elements (rounded rectangles) and tertiary hits (purple and pink circles). (**B**) Proximity maps of c-di-GMP riboswitch aptamer in 10 μM (left) or 0 μM (right) c-di-GMP, folded in 0 mM Mg²⁺, showing absence of tertiary proximities that are present for the aptamer folded in 10 mM Mg²⁺. (**C**) Five MCM models with lowest Rosetta energy of c-di-GMP riboswitch aptamer give an initial visualization of ligand-free ensemble (left); modeling included pseudo-energy constraints from 0 μM c-di-GMP and 10 mM Mg²⁺ proximity map. The crystal structure is shown at right for comparison.

crystallographic studies but later shown to stabilize a K-turn linker between the aptamers (*Figure 5D,E*) (*Kladwang et al., 2012a*; *Esquiaqui et al., 2014*; *Ruff and Strobel, 2014*).

Compared to the riboswitch aptamers above, much less was previously known about the AdoCbl riboswitch aptamer. Indeed, long-standing controversies in modeling this ncRNA raised the possibility of differences in both secondary and tertiary structure in states with and without ligand (*Ravnum and Andersson, 2001*; *Nahvi et al., 2002*; *Vitreschak et al., 2003*; *Nahvi et al., 2004*; *Barrick and Breaker, 2007*; *Miao et al., 2015*). Nevertheless, MCM revealed striking similarities between the states, most clearly visible in MOHCA-seq data (*Figure 6A*). In terms of secondary structure, features for six major helical stacks appeared identical with and without ligand. In terms of tertiary structure, the data without ligand showed reduction but not complete loss of strong tertiary proximities between helices that enclose the ligand-binding site (*Figure 6A*). As above, Rosetta modeling produced an initial visualization of a loose structural ensemble that samples, but does not stably remain in, the ligand-bound structure within the nanometer resolution of the MOHCA-seq method (*Figure 6B* and *Table 2*). Again, by rapidly resolving secondary and tertiary structure, this MCM analysis sets a foundation for further dissection of this RNA's ligand-free globule. As a first example, we found that the sole long-range tertiary interaction in the AdoCbl-binding site (an A-minor contact between J11-10 and helix P6) was necessary for ligand binding but not the global tertiary proximities measured by MOHCA-seq (*Figure 6C,D*). Like the data for the glycine riboswitch aptamers above, these data further suggest that the ligand-free proximities in the AdoCbl riboswitch aptamer arise not from direct tertiary contacts but from conformational preferences encoded in the riboswitch aptamer junctions.

## Solution structure determination of a recently discovered ncRNA

The discovery in recent months of cellular internal ribosome entry site (IRES) elements in the 5′-UTRs of the Hox mRNAs has suggested new roles of structured ncRNAs in mammalian development

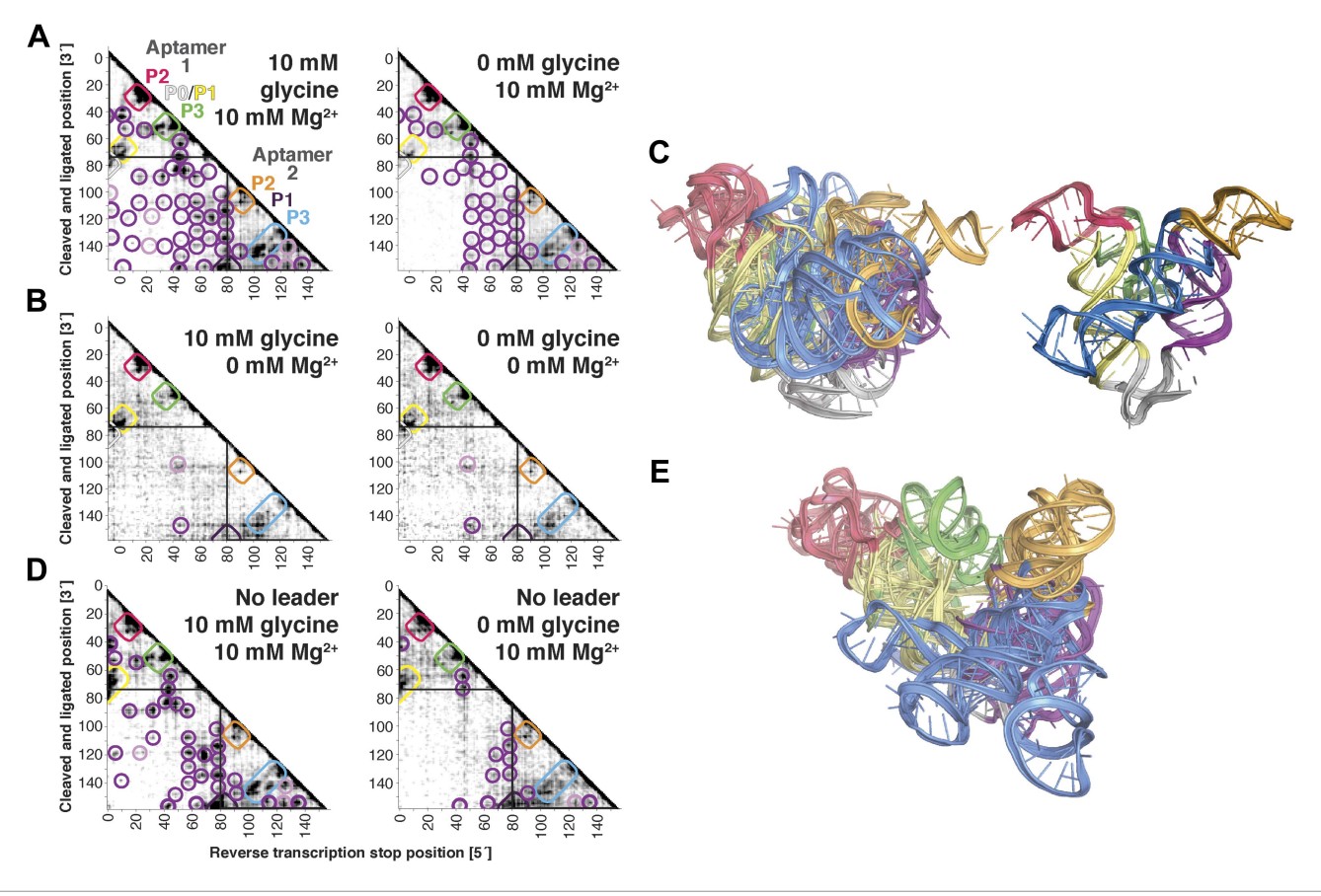

**Figure 5**. The *F. nucleatum* double glycine riboswitch ligand-binding domain retains preformed tertiary structure in its ligand-free state, with a requirement of a leader sequence. (**A**) Proximity maps of glycine riboswitch ligand-binding domain in 10 mM (left) or 0 mM (right) glycine, folded in 10 mM $Mg^{2+}$, with annotated helix elements (rounded rectangles) and tertiary hits (purple and pink circles). A black box encloses each glycine-binding aptamer. (**B**) Proximity maps of glycine riboswitch ligand-binding domain in 10 mM (left) or 0 mM (right) glycine, folded in 0 mM $Mg^{2+}$, showing absence of tertiary proximities both within and between aptamers that are present for the ligand-binding domain folded in 10 mM $Mg^{2+}$. (**C**) Five MCM models with lowest Rosetta energy of glycine riboswitch ligand-binding domain give an initial visualization of ligand-free ensemble (left); modeling included pseudo-energy constraints from 0 mM glycine and 10 mM $Mg^{2+}$ proximity map. The crystal structure, with kink-turn linker grafted from PDB ID 3CC2 (*Kladwang et al., 2012a*), is shown at right for comparison. (**D**) Proximity maps of glycine riboswitch ligand-binding domain without leader sequence in 10 mM (left) or 0 mM (right) glycine, folded in 10 mM $Mg^{2+}$, showing absence of tertiary proximities between aptamers even in presence of $Mg^{2+}$. (**E**) Five MCM models with lowest Rosetta energy of glycine riboswitch ligand-binding domain without leader sequence give an initial visualization of ligand-free ensemble (left); modeling included pseudo-energy constraints from 0 mM glycine and 10 mM $Mg^{2+}$ proximity map.

(*Xue et al., 2015*). To further demonstrate the potential of MCM 3D modeling for investigating systems of timely biological interest, we probed a 176-nt structured domain of the HoxA9 5′-UTR that was shown to be critical for IRES activity. The secondary structure of the domain was determined by $M^2$, along with in vivo mutation-rescue experiments that confirmed key helix elements (*Figure 7A*) (*Xue et al., 2015*). We collected MOHCA-seq data that corroborated the secondary structure described previously and provided independent support for a pseudoknot (pk3-4) connecting two arms of a four-way junction, a common feature of viral IRESes that has not been demonstrated for cellular IRESes (*Figure 7A,B*). We completed the MCM pipeline to computationally model the domain's global 3D structure, revealing a core four-way junction aligned by the pseudoknot, defined at an estimated resolution of 19.5 Å, similar to the loose ensembles seen in our MCM models of ligand-free riboswitch aptamers (*Figure 7C* and *Table 2*). Supporting this global structure and estimated resolution, modeling that used an alternative pk3-4-free secondary structure gave a final conformation within 19.5 Å RMSD of the cluster center with pseudoknot (*Figure 7D*; alternative

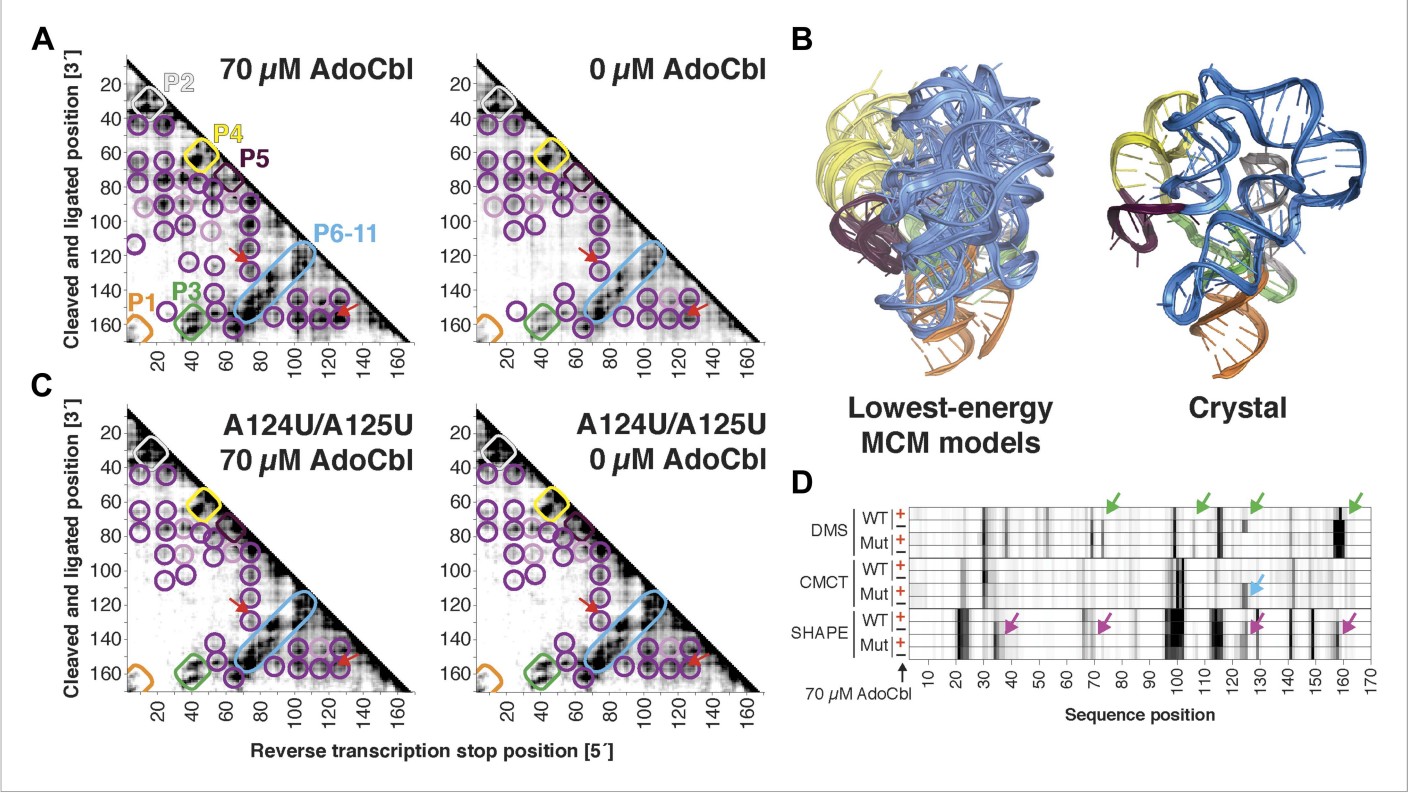

**Figure 6**. The *S. thermophilus* adenosylcobalamin riboswitch aptamer retains preformed tertiary structure without binding ligand. (**A**) Proximity maps of AdoCbl riboswitch aptamer in 70 μM (left) or 0 μM (right) AdoCbl with annotated helix elements (rounded rectangles) and tertiary hits (purple and pink circles). Red arrows indicate regions of proximity between J11-10 and helix P6 in the crystal structure. (**B**) Three MCM models with lowest Rosetta energy of AdoCbl riboswitch aptamer give an initial visualization of ligand-free ensemble (left); modeling included pseudo-energy constraints from 0 μM AdoCbl proximity map. The crystal structure is shown at right for comparison. (**C**–**D**) Mutation of A124 and A125 to U disrupts ligand-binding but not preformed tertiary structure. (**C**) Proximity maps of A124U/A125U mutant AdoCbl riboswitch aptamer in 70 μM (left) or 0 μM (right) AdoCbl, showing retention of tertiary proximity hits similar to the wild-type aptamer in 0 μM AdoCbl. (**D**) 1D chemical mapping reactivities of wild-type and A124U/A125U ('Mut') AdoCbl riboswitch aptamer. Green arrows in DMS reactivity lanes indicate locations of reduced reactivity in the presence of ligand in the wild type only. Residues 124 and 125 are unreactive to DMS in the A124U/A125U mutant because DMS does not modify U residues. Blue arrow in CMCT reactivity lanes indicates lack of protection of U124 and U125 in the mutant in the presence of ligand. Magenta arrows in SHAPE reactivity indicate locations of reduced/altered reactivity in the presence of ligand in the wild type but not in the mutant. All ligand-dependent protections are located in or near the AdoCbl-binding site in the riboswitch aptamer (*Peselis & Serganov, 2012*).

secondary structure from Extended Data *Figure 6C* in ref. [*Xue et al., 2015*]). Given the conformational heterogeneity of this 176-nucleotide ncRNA in the absence of partners, MCM appears to be the only front-line technique able to resolve this domain's global structure in isolation, which we propose to be a recognition site for the mammalian 80S ribosome. These models provide fiducial landmarks to seek in upcoming cryo-electron microscopy studies of the HoxA9 IRES in the presence of the ribosome and other macromolecular partners (PDB-formatted models are available online in *Source code 1*).

## Discussion

Despite rapid advances in secondary structure inference from chemical mapping and computational modeling, inference of three-dimensional ncRNA structures has been bottlenecked by an inability to experimentally determine how a molecule's helices and non-canonical motifs assemble into complex global structures. The MCM methods described herein leverage tabletop sequencing platforms and commercially available reagents to provide a straightforward route to such proximity information. In the MCM pipeline, both $M^2$ and MOHCA-seq experiments provide critical data to guide Rosetta

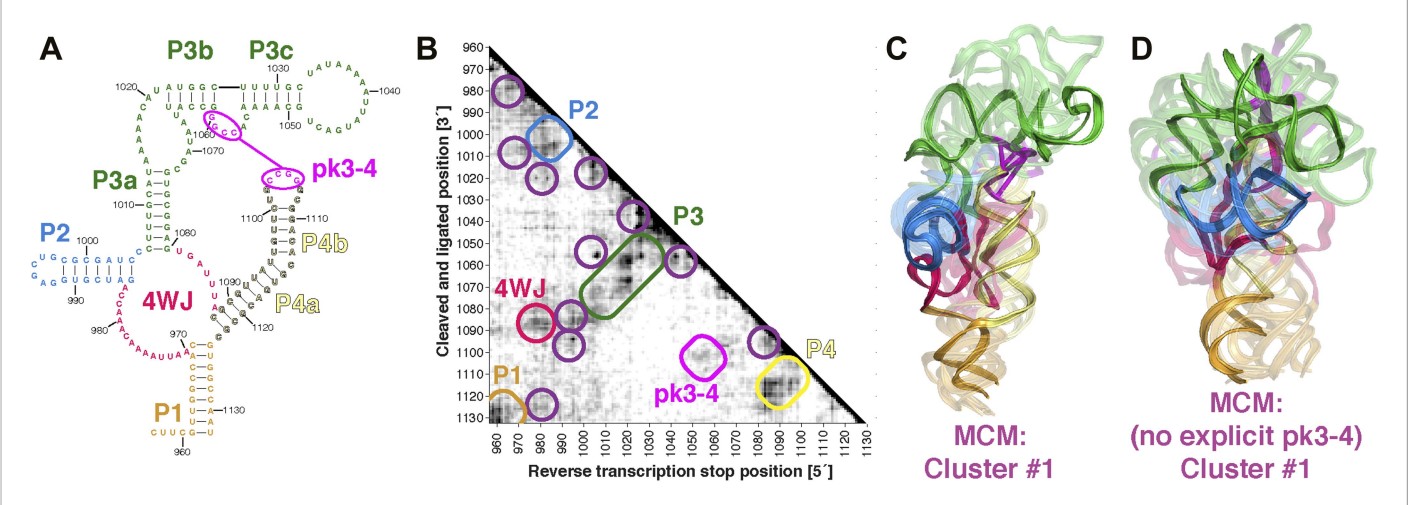

**Figure 7.** MCM analysis of a recently discovered human cellular mRNA IRES domain. (**A**) $M^2$-derived secondary structure model of HoxA9 5′-UTR 957–1132 internal ribosome entry site (IRES) domain. (**B**) MOHCA-seq proximity map of HoxA9 IRES domain, with evidence for secondary structure elements and pseudoknot predicted by $M^2$ and validated by mutate-and-rescue experiments (*Xue et al., 2015*) (rounded rectangles). Tertiary hits are indicated by purple circles. (**C**) MCM model of HoxA9 IRES domain with explicit pseudoknot pk3-4. (**D**) MCM model of HoxA9 IRES domain without explicit pseudoknot pk3-4. In (**C–D**), cluster center (opaque) and four additional models (semi-transparent) from the top cluster are shown. Magenta lines connect base pairs of pk3-4 in each cluster center model. Top cluster models for both modeling setups are available in *Source code 1*.

computational modeling. $M^2$ defines the secondary structure but cannot generally constrain helical elements into accurate tertiary structures, while MOHCA-seq provides tertiary proximities but does not reveal secondary structure as finely as $M^2$. Analogous to how pipelining Heteronuclear Single Quantum Coherence (HSQC) spectroscopy, NOE measurements, and computational modeling allows routine NMR-based structure determination of small RNA motifs, we have shown that the $M^2$/MOHCA-seq/Rosetta pipeline achieves models for larger ncRNA structures, including blind test cases, with accuracies of 1 nm.

Beyond resolving the global structures of well-structured ncRNA states, MCM provides global structural frameworks to guide structure-function analyses of ncRNAs that adopt multiple conformations. MCM data corroborate and refine models of ligand-independent tertiary structure in a c-di-GMP riboswitch aptamer and reveal similarly preformed tertiary structure in glycine and AdoCbl riboswitch aptamers. Mutational analysis confirms the importance of junctions in defining these ligand-free global tertiary preferences. Such preformed structure may be needed for rapid kinetic capture of cognate ligands in vivo. We propose that current exceptions to this rule, such as the thiamine pyrophosphate riboswitch aptamer (*Ali et al., 2010*), may also form such 'loose tertiary globules' when probed with flanking elements or macromolecule partners that have not yet been identified, as they were for the glycine riboswitch aptamers. As such new elements are identified, MCM will provide a uniquely rapid method to test their effects on riboswitch aptamer structure both with and without ligand.

As an example of rapid modeling of a complex and biologically important ncRNA, MCM has revealed a global structure for a HoxA9 IRES domain almost immediately after its discovery as a regulator of vertebrate body plan. The structure is less well-defined than the other ncRNAs but exhibits a pseudoknot that is a common feature of viral IRESes. Other macromolecules that recognize this IRES domain, including the 80S ribosome, may further stabilize or remodel this ncRNA's 'naked' tertiary structure, and MCM will provide a rapid approach to characterizing such structural effects.

In addition to establishing MCM as a routine approach to achieve medium-resolution structures of ncRNA domains, our results suggest both experimental and computational extensions to expand the technology. Cell-permeable nucleotide analogs or chemical modifiers may enable in vivo 3D structure determination. The experimental cost of MCM grows quadratically with the length of the transcript, limiting current analysis to domains of hundreds of nucleotides; however, >1000-nt transcripts may

become feasible as sequencing costs continue to decrease and if new strategies to bypass reverse transcription stops are implemented (*Siegfried et al., 2014*). A fraction of observed MOHCA-seq constraints do not reflect structures solved by crystallography; quantitative relationships correlating MCM data to structural features should further improve modeling accuracy and enable error estimation and cross-validation by leaving out constraints during modeling, analogous to continuing advances in NMR methods. Additionally, stabilization of alternative states by mutation, coupled with ensemble-based analyses, may allow for more detailed dissection of multiple ncRNA tertiary conformations and their functions. Finally, improving computational methods may enable refinement of 1-nm resolution models to higher resolution, possibly near-atomic accuracy; this computational problem appears significantly more amenable than completely de novo 3D modeling. We hope that the public availability of our MCM data, their acquisition in ongoing RNA-puzzles, and the detailed methodological descriptions in this report will help accelerate these developments and more comprehensively illuminate the complex structures underlying ncRNA biology.

# Materials and methods

## Mutate-and-map ($M^2$)

High-confidence secondary structures derived from $M^2$ experiments were used as inputs to computational modeling. We previously collected $M^2$ data on the AdoCbl riboswitch aptamer in the presence of 60 µM AdoCbl and separately performed $M^2$ on the class I ligase using experimental methods and data analysis that have been described in detail (*Kladwang et al., 2011a*; *Cordero et al., 2014*). The $M^2$ data are available on the RNA Mapping Database (RMDB) with accession IDs RNAPZ6_1M7_0002 (AdoCbl riboswitch aptamer) and CL1LIG_1M7_0001 (class I ligase). The solution conditions were 50 mM Na-HEPES, pH 8.0, and 10 mM $MgCl_2$, as used previously and for MOHCA-seq experiments below. *Figure 3—figure supplements 1, 2, 7* present previously unpublished $M^2$ data and analysis for the AdoCbl riboswitch aptamer, class I ligase, and lariat-capping ribozyme, respectively.

## MOHCA-seq measurements

### RNA preparation

Double-stranded DNA templates for the RNAs of interest were constructed using PCR assembly with primers purchased from IDT (Integrated DNA Technologies, San Diego, CA) (*Kladwang et al., 2011b*). RNAs were transcribed at 37°C for 3 hr in 320 µl reactions containing 32 pmol of dsDNA template, 100 mM Tris–HCl, pH 8.1, 200 mM $MgCl_2$, 3.5 mM spermidine, 0.1% Triton X-100, 40 mM DTT, 4% PEG 8000, 20 U T7 RNA Polymerase (New England Biolabs [NEB], Ipswich, MA), 1 mM NTPs, and 0.5 mM 2′-$NH_2$-2′-deoxy-ATP (TriLink BioTechnologies, San Diego, CA). The yield of transcription was 1–1.5 nmol after purification using RNA Clean & Concentrator columns (Zymo Research, Irvine, CA). Purified RNA was then ethanol precipitated to concentrate it for 5′-end labeling using a 5′ EndTag kit and fluorescein maleimide (Vector Labs, Burlingame, CA) to enable gel visualization in subsequent steps; the RNA was then purified again using RNA Clean & Concentrator columns. The hydroxyl radical source, isothiocyanobenzyl-EDTA chelating Fe(III) (ITCB-Fe(III)•EDTA) (Dojindo Molecular Technologies, Inc., Santa Clara, CA), was covalently attached to the 2′-$NH_2$ groups on the RNA backbone using a two-step process. First, to couple ITCB-EDTA to the RNA, 0.5 mg ITCB-EDTA was dissolved in 5.7 µl 0.4 M $KPO_4$, pH 8.5 for a concentration of 200 mM, then mixed with 15 µl of RNA and 15 µl 0.8 M $KPO_4$, pH 8.5. The coupling reactions were incubated at 37°C for 12–16 hr. Then, 1/3 vol of 200 mM $FeCl_3$ was added and the coupling reactions were incubated at room temperature for 15 min, after which 10 µl 500 mM Na-EDTA, pH 8.0 was added to chelate excess Fe(III). After purifying with RNA Clean & Concentrator columns to remove excess reagents, the RNA was PAGE-purified using denaturing 8% polyacrylamide/7 M urea gels. Bands were located by scanning with a Typhoon imager (GE, Pittsburgh, PA) for fluorescein fluorescence, and excised gel slices were immersed in approximately 500 µl RNase-free $H_2O$ in non-stick tubes overnight at 4°C to elute the RNA. The RNA was purified from the eluate using RNA Clean & Concentrator columns and stored at −20°C. The yield of PAGE purification was 10–30% of the amount loaded on the gel, which was sufficient for multiple MOHCA-seq experiments.

## Activation of radical source

Before activating the radical source to produce spatially localized hydroxyl radicals, 3–4 pmol of folded RNA was prepared in 50 mM Na-HEPES, pH 8.0, and 10 mM MgCl$_2$, as follows. First, the RNA was heated to 65°C in HEPES buffer in 8 µl for 3 min, then cooled to room temperature for 10 min; then 1 µl of 100 mM MgCl$_2$ was added and the RNA was heated to 50°C for 5 min, then cooled to room temperature for 10 min. (For P4–P6, HEPES buffer and MgCl$_2$ were added concurrently, and the RNA was incubated for 10 min at room temperature. For ligand-bound cyclic-di-GMP riboswitch aptamer samples, 100 pmol cyclic-di-GMP [BioLog, Bremen, Germany] was added before the first heating step for 10 µM in a 10 µl reaction. For ligand-bound glycine riboswitch ligand-binding domain samples, 100 nmol glycine [Sigma, St. Louis, MO] was added before the first heating step for 10 mM in a 10 µl reaction. For ligand-bound adenosylcobalamin [AdoCbl] riboswitch aptamer samples, 700 pmol AdoCbl [Sigma] was added before the first heating step for 70 µM in a 10 µl reaction, and all steps until post-fragmentation ethanol precipitation [below] were performed under low-light conditions. For 0 mM Mg$^{2+}$ conditions, RNase-free H$_2$O was added instead of MgCl$_2$.) After folding, the radical source was activated by adding 1 µl 100 mM sodium ascorbate stock for a final concentration of 10 mM. A control reaction was also prepared with 1 µl RNase-free H$_2$O added instead of ascorbate; this reaction was carried through all subsequent steps in parallel. After 5 to 30 min of incubation at room temperature (10 min was standard), 1 µl 100 mM thiourea was added to quench the Fenton reaction. The RNA fragments were ethanol precipitated as follows: First, 1 µl GlycoBlue (Life Technologies, Carlsbad, CA) and 1/10 vol (1.2 µl) of 3 M sodium acetate, pH 5.2 were added to the quenched reaction and mixed. Then, 3 vol (40 µl) of 100% ethanol chilled on dry ice were added, and the sample was mixed. Samples were spun down immediately in a tabletop microcentrifuge at maximum speed for 20 min, washed with 50 µl ice-cold 70% ethanol, then spun down again at max speed for 10 min. The supernatant was removed by pipetting and the pellets were allowed to dry, after which 15 µl RNase-free H$_2$O was added and the pellet was allowed to dissolve for 10 min at room temperature.

## Repair of 3′ ends

To remove 3′-phosphates left by hydroxyl radical strand scission events in the RNA backbone (*Balasubramanian et al., 1998*; *Pogozelski and Tullius, 1998*), the purified RNA fragments were treated with T4 polynucleotide kinase (T4 PNK) in conditions that promoted 3′-phosphatase activity (*Cameron and Uhlenbeck, 1977*). Each end-repair reaction contained 14 µl of the RNA from the previous step (the remaining 1 µl was used for quality checks by capillary electrophoresis), as well as 50 mM Na-MES, pH 6.0, 10 mM MgCl$_2$, 5 mM DTT, 5 µM ATP, and 10 units T4 PNK (NEB), in a 20 µl reaction volume and was incubated at 37°C for 30 min. After end-repair, RNAs were ethanol precipitated as above, except with 0.5 µl GlycoBlue, 2.05 µl 3 M sodium acetate, pH 5.2, and 68 µl 100% ethanol chilled on dry ice. The dried pellets were resuspended in 5.2 µl RNase-free H$_2$O. The T4 PNK reaction was necessary and sufficient for ligation of a pre-adenylated ssDNA universal adapter sequence to the radical-cleaved fragments (see next section), based on experiments with PAGE-purified RNA fragments cleaved by hydroxyl radicals from FeEDTA in solution. The ligation efficiency after T4 PNK treatment was ∼50%, suggesting that non-phosphate end products of strand scission, such as 3′-phosphoglycolates, may not be removed by T4 PNK; however, the yield of ligated product was still sufficient for reverse transcription using limiting amounts of sequencing primers.

## Ligation of ssDNA universal adapter sequence

To prepare the fragmented RNA for reverse transcription, a pre-adenylated and 3′-blocked ssDNA universal adapter sequence (Universal miRNA cloning linker, NEB) was ligated to the 3′-end of the end-repaired fragments. Each ligation reaction contained 4.9 µl of the RNA from the previous step (the remaining 0.3 µl was used for quality checks by CE), as well as 1× T4 RNA ligase buffer (NEB), 15% PEG 8000, 4–5 pmol ssDNA adapter, and 200 U T4 RNA ligase 2 truncated, K227Q or KQ mutant (NEB), in a 10 µl reaction volume. The reactions were incubated at 4°C for 12 hr, followed by heat inactivation at 65°C for 20 min. After ligation, reaction volumes were supplemented with 50 µl RNase-free H$_2$O, and RNAs were purified from the reaction using RNA Clean & Concentrator columns, which also removed excess ssDNA adapter. Ligation was assayed using 5% of the purified sample by observing the mobility shift of the full-length RNA on CE. Mobility shifts of radical-cleaved fragments were not directly observed in this quality check but were observed in pilot experiments, as described above.

## Reverse transcription with sequencing primers (first adapter)

After ligation of the ssDNA adapter, RNAs were reverse transcribed with sequencing primers containing, in 5′ to 3′ order, a 5′-fluorescein modification, the Illumina TruSeq Universal adapter, 12-nucleotide barcodes (sequence-balanced in sets of 4 primers), and a primer for the ssDNA adapter sequence on the 3′-end (sequence listed in *Supplementary file 1*). Three of the four primers were used for reverse transcription of the ascorbate-treated sample and the remaining primer was used for reverse transcription of the no-ascorbate control sample. Each 15 μl reverse transcription reaction contained 250 fmol sequencing primer (which is limiting compared to the ~2–3 pmol of RNA remaining at this stage), 1× First Strand buffer (Life Technologies), 5 mM DTT, 0.8 mM dNTPs, and 120 U SuperScript III (Life Technologies) and was incubated at 55°C for 30 min. To degrade the RNA templates, 5 μl of 0.4 M NaOH was added and the samples were incubated at 90°C for 3 min. After cooling on ice for 3 min, the cDNA samples were neutralized by addition of 1 μl of an acid quench (2 ml 5 M NaCl, 2 ml 2 M HCl, and 3 ml 3 M Na-acetate) and then purified by incubating with DynaBeads magnetic beads (Life Technologies) conjugated to double-biotin-labeled ssDNA complementary to the TruSeq adapter. For the glycine riboswitch ligand-binding domain with K-turn construct, the cDNA samples after RNA hydrolysis and neutralization were split in half. One half was purified as described above, and the other was purified using AMPure XP beads (Beckman Coulter, Indianapolis, IN), to provide size-selection for larger fragments corresponding to tertiary proximities between more distant sequence positions. Approximately, 65–75% of the sequencing primer was extended in the reverse transcription reactions, based on CE detection and quantification of 'null ligation' reads (consisting of the second sequencing adapter ligated directly to the sequencing primer) after paired-end sequencing.

## Ligation of second sequencing adapter

The second sequencing adapter ("Second ligation adapter 1", listed in *Supplementary file 1*), derived from the TruSeq Indexed adapter with a 5′-phosphate (to enable ligation to the cDNA) and a 3′-phosphate (to block circularization), was ligated onto the 3′-ends of the cDNAs. The four cDNA samples for each RNA/condition were pooled prior to the ligation reaction. Each 50 μl ligation reaction contained 1.25 μM adapter, 1× CircLigase buffer (Epicentre, Madison, WI), 50 μM ATP, 2.5 mM $MnCl_2$, 4% PEG 1500, and 250 U CircLigase I (Epicentre) and was incubated at 68°C for 2 hr, followed by heat inactivation at 80°C for 10 min. The samples were purified by DynaBeads as above with magnetic separation, and a fraction of the sample was run on a capillary electrophoresis machine (Applied Biosystems, Foster City, CA) with co-loaded fluorescein-labeled standards and analyzed using HiTRACE (*Yoon et al., 2011*) to estimate the concentration of ligated cDNA by comparing the total intensities of ligated cDNAs to the total intensity of the standard. We observed approximately 50% ligation yield from the CircLigase reaction, consistent with earlier work (*Seetin et al., 2014*). For the glycine riboswitch ligand-binding domain with K-turn construct, the DynaBead-purified samples were ligated with "Second ligation adapter 1" as described above, while the AMPure XP-purified samples were ligated with a separate sequencing adapter ("Second ligation adapter 2", listed in *Supplementary file 1*). This allowed multiplexed sequencing of the DynaBead- and AMPure XP-purified sample libraries.

## Sequencing

The MOHCA-seq cDNA libraries were sequenced using either 50-cycle MiSeq v2 kits or 150-cycle MiSeq v3 kits on Illumina MiSeq instruments or using an Illumina HiSeq 2500 instrument (Elim Biopharmaceuticals, Hayward, CA). For runs using MiSeq v2 kits, the following protocol was used to prepare the library for sequencing: beads harboring 25 fmol of single-stranded library fragments, usually about 1/5 of the final library volume after ligation of the second adapter, were mixed with 2.5 fmol of PhiX dsDNA control (Illumina, Inc., Hayward, CA) and EB buffer (Qiagen, Venlo, Netherlands) to 5 μl total. Then, 5 μl of 0.2 N NaOH was added and the fragments were eluted for 10 min at room temperature. The supernatant of the magnetic beads was diluted into chilled HT1 buffer (10 μl added to 990 μl HT1), and then diluted again (375 μl added to 225 μl HT1) before loading all 600 μl onto MiSeq kits following manufacturer instructions. Paired-end sequencing involved 51 and 25 sequencing cycles for the first and second reads, respectively. For runs using MiSeq v3 kits, the following protocol was used to prepare the library for sequencing: beads harboring 32 fmol of single-stranded library fragments were mixed with 4.2 μl of 95% formamide and 10 mM EDTA and heated at

90°C for 2 min and 40 s, then cooled at room temperature for several minutes. The beads were magnetically separated, and 4 fmol of PhiX dsDNA control (Illumina) was added to the isolated supernatant. Then, 5 µl of 0.2 N NaOH was added and the samples were denatured for 10 min at room temperature. This sample was diluted into chilled HT1 buffer (10 µl added to 990 µl HT1) and 600 µl of this dilution was loaded onto MiSeq kits following manufacturer instructions. Paired-end sequencing involved 51 and 51 sequencing cycles for the first and second reads, respectively.

The FASTQ data generated by sequencing were processed with the MAPseeker v1.2 software (accessible through the RMDB server at http://rmdb.stanford.edu/tools/), giving final text files in RDAT format corresponding to the number of fragments observed for each pair of possible cleavage and oxidation position (corresponding to 5′ and 3′ positions of cDNA fragments). Our sequencing runs generally produced around 15–20 million total reads, of which around 60–80% contained the ssDNA adapter RT priming site and aligned to the RNA sequence in read 1, and 25–35% aligned to the RNA sequence in both read 1 and read 2. Approximately, 25% of aligned reads corresponded to control reactions with radical sources but without ascorbate treatment, while 75% of aligned reads corresponded to the ascorbate-treated, fragmented RNA, as expected from our method of dividing the ascorbate-treated sample into three fractions for reverse transcription. 25–35% of reads containing the RT priming site were null ligations of unextended reverse transcription primer to the second sequencing adapter.

For the glycine riboswitch ligand-binding domain with leader sequence, the raw data for the DynaBead-purified and AMPure XP-purified (size-selected) libraries were combined as follows: the sum of the signal at each sequence separation (along the diagonals of the 2D map) was calculated for each data set. For sequence separations between 50 and 100, the signal was averaged across consecutive bins of 2; above 100, the signal was averaged across bins of 10. Then, to account for size-selection by AMPure XP purification, the ratio of AMPure XP signal to DynaBead signal at each sequence separation was calculated and the AMPure XP signal was 'rebalanced' by multiplying it by the maximum AMPure XP/DynaBead ratio and then dividing by AMPure XP/DynaBead ratio at that sequence separation. Finally, the rebalanced AMPure XP data set was combined with the DynaBead data set by taking the mean (weighted by the inverse error squared) between the data sets. The raw data have been deposited in the RMDB with the following accession IDs:

P4-P6: TRP4P6_MCA_0001-0004
c-di-GMP riboswitch aptamer with 10 µM c-di-GMP: CDIGMP_MCA_0002
c-di-GMP riboswitch aptamer with 0 µM c-di-GMP: CDIGMP_MCA_0003
c-di-GMP riboswitch aptamer with 10 µM c-di-GMP and 0 mM $Mg^{2+}$: CDIGMP_MCA_0006
c-di-GMP riboswitch aptamer with 0 µM c-di-GMP and 0 mM $Mg^{2+}$: CDIGMP_MCA_0007
Glycine riboswitch ligand-binding domain with 10 mM glycine: GLYCFN_MCA_0002
Glycine riboswitch ligand-binding domain with 0 mM glycine: GLYCFN_MCA_0003
Glycine riboswitch ligand-binding domain including leader, with 10 mM glycine: GLYCFN_KNK_0005
Glycine riboswitch ligand-binding domain including leader, with 0 mM glycine: GLYCFN_KNK_0006
Glycine riboswitch ligand-binding domain including leader, with 10 mM glycine and 0 mM $Mg^{2+}$: GLYCFN_KNK_0009
Glycine riboswitch ligand-binding domain including leader, with 0 mM glycine and 0 mM $Mg^{2+}$: GLYCFN_KNK_0010
AdoCbl riboswitch aptamer with 70 µM AdoCbl: RNAPZ6_MCA_0002
AdoCbl riboswitch aptamer with 0 µM AdoCbl: RNAPZ6_MCA_0003
AdoCbl riboswitch aptamer (A124U/A125U mutant) with 70 µM AdoCbl: RNAPZ6_MCA_0006
AdoCbl riboswitch aptamer (A124U/A125U mutant) with 0 µM AdoCbl: RNAPZ6_MCA_0007
Class I ligase: CL1LIG_MCA_0001-0003
Lariat-capping ribozyme: RNAPZ5_MCA_0001-0002
HoxA9 5′-UTR 957–1132 domain: HOXA9D_MCA_0001

## Quantification of oxidative damage

To evaluate the rates of strand scission and reverse transcription-terminating modifications, we quantified data from both paired-end sequencing and CE for the model hairpin RNA shown in *Figure 2B*, which contained a single radical source at U13 (blue and orange bars, *Figure 2D*). As a background control, we quantified data collected in parallel on a sample that did not have a tethered radical source but was otherwise treated identically (gray bars, *Figure 2D*).

To quantify strand scission events from sequencing data, we calculated the sum of all counts for fragments that shared the same ssDNA adapter ligation site, giving a projection of the raw counts after MAPseeker analysis (*Figure 2D*, top center). We then normalized the counts at each position by the number of counts for all fragments that were ligated at the 3′ end of the RNA, giving the fraction modified at each position. For an independent measurement of strand scission, we also ran a fraction of the same sample on CE, after radical source activation but before end-repair, and quantified the observed 5′-end labeled cleavage products using HiTRACE (*Yoon et al., 2011*) (*Figure 2D*, top). One residue position (marked by *, *Figure 2D*) overlapped with a peak corresponding to bleed-through fluorescence from the reference channel, so the quantification of strand scission at that position is uncertain. The strand scission profiles from sequencing and CE are similar, showing fragmentation both near the position of the radical source and on the strand of the helix complementary to the radical source (*Figure 2D*, top and top center). We calculated the rate of strand scission from the CE and sequencing data by taking the sum of the fraction modified over all internal cleavage sites, excluding the reference bleed-through peak from CE and a high-background peak from the sequencing data (marked by *, *Figure 2D*). The rate of strand scission was similar between the CE and sequencing data (*Figure 2E*).

To quantify reverse transcription stops from sequencing data, we calculated the sum of all raw counts for fragments with stops at the same position after MAPseeker analysis. To account for reverse transcriptase attenuation, we normalized the signal at each position by the total counts for all fragments that were ligated 3′ of that position and were terminated at that position or 5′ of that position, analogous to the exact correction applied for CE data (*Figure 2D*, bottom) (*Yoon et al., 2011*). For an independent measurement of reverse transcription stops, we also ran a fraction of the same sample on CE, after reverse transcription with sequencing primers but before ligation of the second sequencing adapter, and quantified the reactivity of each position using HiTRACE (*Yoon et al., 2011*) (*Figure 2D*, bottom center). The reverse transcription stop profiles from sequencing and CE are similar, showing reverse transcription-terminating events on the strand of the helix complementary to the radical source, as well as a stronger termination event at the location of the radical source (*Figure 2D*, bottom center and bottom). We calculated the rate of reverse transcription stops from the CE and sequencing data by taking the sum of the fraction modified over all stop sites, excluding the signal at residue 13 because the reactivity at that position is due to the tethered radical source and not oxidative modifications. The rate of RT stops was similar between the CE and sequencing data and was at least sixfold higher than the rate of strand scission (*Figure 2E*).

## MOHCA-seq data analysis
### General MOHCA-seq analysis framework
Analysis of MOHCA-seq data requires modeling backgrounds, modulation from reverse transcription attenuation, and sources of error. Here, we outline a general analysis framework for MOHCA-seq experiments, with the following section describing a statistical procedure developed to reach a numerical solution, that gave consistent proximity maps. The resulting processed proximity maps have been deposited in the RMDB with the following accession IDs:

P4–P6: TRP4P6_MCA_0000
c-di-GMP riboswitch aptamer with 10 µM c-di-GMP: CDIGMP_MCA_0000
c-di-GMP riboswitch aptamer with 0 µM c-di-GMP: CDIGMP_MCA_0001
c-di-GMP riboswitch aptamer with 10 µM c-di-GMP and 0 mM Mg$^{2+}$: CDIGMP_MCA_0004
c-di-GMP riboswitch aptamer with 0 µM c-di-GMP and 0 mM Mg$^{2+}$: CDIGMP_MCA_0005
Glycine riboswitch ligand-binding domain with 10 mM glycine: GLYCFN_MCA_0000
Glycine riboswitch ligand-binding domain with 0 mM glycine: GLYCFN_MCA_0001
Glycine riboswitch ligand-binding domain including leader, with 10 mM glycine: GLYCFN_KNK_0003
Glycine riboswitch ligand-binding domain including leader, with 0 mM glycine: GLYCFN_KNK_0004
Glycine riboswitch ligand-binding domain including leader, with 10 mM glycine and 0 mM Mg$^{2+}$: GLYCFN_KNK_0007
Glycine riboswitch ligand-binding domain including leader, with 0 mM glycine and 0 mM Mg$^{2+}$: GLYCFN_KNK_0008
AdoCbl riboswitch aptamer with 70 µM AdoCbl: RNAPZ6_MCA_0000
AdoCbl riboswitch aptamer with 0 µM AdoCbl: RNAPZ6_MCA_0001

AdoCbl riboswitch aptamer (A124U/A125U mutant) with 70 μM AdoCbl: RNAPZ6_MCA_0004
AdoCbl riboswitch aptamer (A124U/A125U mutant) with 0 μM AdoCbl: RNAPZ6_MCA_0005
Class I ligase: CL1LIG_MCA_0000
Lariat-capping ribozyme: RNAPZ5_MCA_0000
HoxA9 5′-UTR 957–1132 domain: HOXA9D_MCA_0000

A MOHCA-seq product stemming from a radical cleavage event at nucleotide $j$ and a radical source at nucleotide $i$ corresponds to a sequence $i + 1$ to $j − 1$ ligated between the two adapters necessary for paired-end Illumina sequencing. The frequency $F_{ij}$ of such products is related to the proximity of $i$ and $j$ but is modulated by the actual distribution of radical sources, for example, primarily at adenosines for 2′-NH$_2$-dATP-incorporating transcripts. The frequency is also suppressed by signal attenuation for long sequence separations $j − i$ due to the possibility of reverse transcription termination between $i$ and $j$. A master expression for these MOHCA-seq frequencies is:

$$F_{ij} = \sum_s p_i^s \left(1 - p_{i+1}^s\right)\left(1 - p_{i+2}^s\right)\ldots\left(1 - p_{j-1}^s\right)q_j^s \varepsilon(s), \quad (1)$$

with $s$ indexing the possible source positions (0, 1, … $N$ with $s = 0$ corresponding to no source) and $\varepsilon(s)$ the fraction of transcripts containing a source at $s$. Values $p_i^s$ give the probability that a reverse-transcription terminating event occurs at the nucleotide immediately 3′ to $i$ for a transcript with source at $s$, and values $q_j^s$ give the probability of a cleavage event between $j − 1$ and $j$ for a transcript with source at $s$. This expression simplifies in the limit of no background processes and low-oxidative damage rates (including radical-induced strand scission and nucleobase oxidation) ($p_i^s, q_j^s \ll 1$, except at $p_i^{s=i} = 1$, corresponding to a reverse transcriptase stop at the site of radical source attachment). In that limit, *Equation 1* reduces to:

$$F_{ij} = q_j^i \varepsilon(i), \quad (2)$$

That is, the observed frequencies provide a direct readout of the probability that a source at $i$ leads to radical cleavage at position $j$. This was the limit assumed in prior gel-based MOHCA analysis based on reading out $i$ through cleavage of phosphorothioate tags associated with the radical sources. In the present MOHCA-seq protocol, we found empirically that the reverse transcription readout of $i$ led to a more complete portrait of the proximity map, including information at nucleotides $i$ at which radical sources were not attached. Indeed, increasing the incorporation rate of radical sources or the time of ascorbate-induced radical damage produced higher signal-to-noise data sets with clear proximity map signals, despite bringing the analysis away from the regime of single-hit incorporation and damage (*Figure 2—figure supplements 1, 2*). The extra information was derived from oxidative damage events that did not lead to backbone cleavage encapsulated in the term $p_i^s$ (*Figure 2B–E*; also, compare *Figure 1D* to initial MOHCA-gel data [*Das et al., 2008*], these events were previously invisible to gel-based analysis), but required a more advanced analysis to elicit the signal from the raw data.

In the general case, the number of observed frequencies $F_{ij}$ is on the order of $N(N − 1)/2$, whereas the total number of model parameters $\varepsilon(s)$, $p_i^s$, and $q_j^s$ is substantially higher ($>2N^2$), leading to an ill-posed problem. However, basic chemical considerations reduce the number of parameters. First, we assume that the cleavage fraction that allows ssDNA adapter ligation, $q_j^s$, is composed of a 'background' rate of cleavage $b_j$ that is independent of source $s$ but can vary with nucleotide $j$ (e.g., due to inline attack during the fragmentation reaction), and additional, correlated cleavage due to hydroxyl radical-induced strand scission events (the desired MOHCA-seq signal), parameterized by $\pi_j^s$:

$$\text{Cleavage at } j \text{ from source } s = q_j^s = 1 - \left(1 - b_j\right)\left(1 - \pi_j^s\right) \approx b_j + \pi_j^s. \quad (3)$$

Second, we assume that the stopping events $p_i^s$ are composed of two components: (1) a background rate $r_i$ of uncorrelated oxidative damage (e.g., due to hydroxyl radicals from FeEDTA that has self-cleaved from the RNA during ascorbate treatment, which would provide background similar to standard FeEDTA hydroxyl radical footprinting) and (2) additional, correlated oxidative damage due to hydroxyl radicals generated at source $s$, which is tethered to the RNA (the desired MOHCA-seq signal), $\pi_i^s$:

$$\text{Stop at } i \text{ from source } s = p_i^s = 1 - \left(1 - r_i\right)\left(1 - \rho_i \pi_i^s\right) \approx r_i + \rho_i \pi_i^s. \quad (4)$$

In *Equations 3, 4*, a reduction in the number of parameters arises from assuming that oxidative damage events producing reverse transcription stops occur at rates proportional to backbone cleavage rates by a factor $\rho_i$. That is, $\pi_i^s$ parameterizes the local effective concentration of radicals at $i$ from source $s$, and the partitioning of these radicals into events that lead to cleavage (contributing to $q_i^s$) vs total damage that can terminate reverse transcription (contributing to $p_i^s$) is dependent on the chemical environment of the site $i$ and not on source $s$. The total number of parameters thus reduces from greater than 2N² to $N(N-1)/2$ for $\pi_i^s$ and $3N$ for $b_j$, $r_i$, and $\varepsilon(s)$. By further enforcing positivity of each of these parameters and assuming that $\pi_i^s$ is sparse, that is, that each nucleotide gives non-negligible cleavage at a number of residues smaller than $N$, the number of parameters is reduced to below the number of observables, and the problem becomes well-posed. Requiring sparsity of the proximity map is similar to assumptions used in solvent flattening and other density modification approaches in crystallography (*Wang, 1985*).

Determining proximity information from raw observables still requires solving a complex system of non-linear equations. We found that direct least-squares optimization of the thousands of variables $\pi_i^s$ to fit the observed $F_{ij}$, including Laplace priors to enforce sparsity, required hours even with state-of-the-art numerical optimizers. We instead developed a rapid, iterative strategy to carry out the solution, as described next.

## COHCOA

A closure-based •OH correlation analysis (COHCOA) estimates a solution to *Equation 1* to determine a two-point correlation function that is directly read out by MOHCA-seq. In the limit that the fraction damaged at any single nucleotide is smaller than one,

$$
\begin{aligned}
F_{ij} &\approx \sum_s \left[ p_i^s q_j^s - \sum_{i<m<j} p_i^s p_m^s q_j^s \right] \varepsilon(s) \\
&= \langle p_i q_j \rangle - \sum_{i<m<j} \langle p_i p_m q_j \rangle,
\end{aligned}
\tag{5}
$$

where the bracket notation refers to a summation over sources:

$$
\langle f \rangle = \sum_s \varepsilon(s) f^s,
\tag{6}
$$

and terms beyond second order are neglected. The effects of the radical cleavage result in a one-dimensional damage profile $R_i$ (corresponding to uncorrelated oxidative damage) and cleavage profile $B_j$ and a two-point correlation function $Q_{ij}$ (corresponding to correlated oxidative damage or radical source locations), which encodes the desired proximity map:

$$
\begin{aligned}
B_j &= b_j + \langle \pi_j \rangle \\
R_i &= r_i + \rho_i \langle \pi_i \rangle \\
Q_{ij} &= \rho_i \left[ \langle \pi_i \pi_j \rangle - \langle \pi_i \rangle \langle \pi_j \rangle \right],
\end{aligned}
\tag{7}
$$

leading to the equation

$$
F_{ij} \approx R_i B_j + Q_{ij} - \sum_{i<m<j} \left( R_i R_m B_j + R_m Q_{ij} + R_i Q_{mj} + \rho_m Q_{im} B_j \right),
\tag{8}
$$

In the derivation of (8), we have dropped higher order terms corresponding to neglecting higher cumulants of the damage function (e.g., the three-point cumulant $J_{imj} = \langle \pi_i \pi_m \pi_j \rangle - \langle \pi_i \pi_m \rangle \langle \pi_j \rangle - \langle \pi_i \pi_j \rangle \langle \pi_m \rangle - \langle \pi_j \pi_m \rangle \langle \pi_i \rangle + 2\langle \pi_i \rangle \langle \pi_j \rangle \langle \pi_m \rangle$) to match the lowest order assumed in *Equation 5*. We also note that stops due to the possibility of more than one radical source attached to the transcript (neglected in the derivation above) can be modeled accurately, to first order, by including a rate $\varepsilon(i)$ within the general 'background' stopping rate $R_i$ (not shown). In general, all processes that lead to stops or cleavage across all transcripts are subsumed into $R_i$ and $B_j$. As a corollary to this simplification, however, this framework does not seek or enable deconvolution of the separate contributions to each of these background rates.

In practice, *Equation 8* can give unphysical negative values if the subtracted summand becomes large. Following a strategy used in, for example, reference interaction site models for solvation, we 'close' the expansion by solving an equation system that is equal to (8) at lowest order, exact in certain limits, and guaranteed to give positive results:

$$F_{ij} = A_{ij}(R_i B_j + Q_{ij}),  \tag{9a}$$

$$A_{ij} = \prod_{i<m<j}(1 - R_m)\left(1 - \frac{R_i Q_{mj}}{F_{ij}}\right)\left(1 - \frac{\rho Q_{im} B_j}{F_{ij}}\right).  \tag{9b}$$

Intuitively, many of the observed products $F_{ij}$ are due to uncorrelated cleavage events $B_j$ and reverse transcription stop events $R_i$, which produce a 'plaid' background pattern. On top of this background is the desired two-point correlation signal $Q_{ij}$, which is non-zero only when nucleotides $i$ and $j$ have both been chemically modified by the same proximal source. Modulating these signals is an attenuation factor $A_{ij}$, which parameterizes the loss of signal, as a reverse transcriptase must polymerize from $j$ back to $i$. This factor depends on the general background stop rate $R_i$, but also includes two additional terms representing the possibility for additional reverse-transcription-stopping damage correlated with the observed cleavage event at $j$ and the observed stopping event at $i$. (For simplicity and based on separate experiments, we fixed $\rho$, the ratio of chemical modification to backbone cleavage, as a constant at 2.5; changing this value from 1 to 5 gave indistinguishable results.) We note that the *Equation 9a,b* is exact in the case of negligible $Q_{ij}$; see references (*Aviran et al., 2011*; *Kladwang et al., 2011b*; *Kladwang et al., 2014*).

The *Equations 9* are solved through iteration in a single script cohcoa_classic.m available in the MAPseeker package. A starting estimate of $R_i$ and attenuation matrix $A_{ij}$ comes from data corresponding to cleavages in the 3′-flanking region, which is initially assumed to not give specific contacts with the target RNA domain. One-dimensional 'background' profiles $R_i$ and $B_j$ are determined, which best fit the attenuation-corrected data $F_{ij}/A_{ij}$. Subtracting the resulting background matrix $R_i B_j$ from observed $F_{ij}$ results in an initial solution for $Q_{ij}$ via *equation (9a)*. Any point of $Q_{ij}$ that is negative is reset to zero, and these $R_i$, $B_j$, and $Q_{ij}$ give an updated solution for the attenuation matrix $A_{ij}$ via *equation (9b)*. New estimates of $R_i$ and $B_j$ are derived from fitting $F_{ij}/A_{ij} - Q_{ij}$ via equation (9d) and the process is iterated until convergence. This procedure does not require assuming symmetry of the two-point correlation function $Q_{ij}$, but in the end returns an estimate of this matrix only for $(i < j)$, where there are data $F_{ij}$. Propagating Poisson counting errors on $F_{ij}$ in *Equation 9b* gives standard errors on $A_{ij}$, and combining these errors in quadrature with the errors on $F_{ij}$ in *Equation 9a* gives final error estimates for the two-point correlation function $Q_{ij}$. Empirically, 20 or fewer iteration cycles lead to convergence for all data sets tested; 40 iterations have been used in this study to ensure convergence of final $Q_{ij}$ values within 1% (taking less than 1 min on a MacBook Pro 2.8 GHz Intel Core i7 running MATLAB 2012B). A comparison of raw counts and COHCOA-analyzed data is shown in *Figure 1—figure supplement 1*.

For visualization of the COHCOA-analyzed data as 2D proximity maps, we first applied a filter to remove points with signal-to-noise ratio < 1, then applied a 2D smoothing algorithm to aid visualization of the strongest features. Finally, we calculated the mean of the filtered and smoothed data with all negative values set to 0, then scaled the data by dividing by 5 times this value.

## One-dimensional chemical mapping of AdoCbl riboswitch aptamer

We performed one-dimensional (1D) chemical mapping of wild-type AdoCbl riboswitch aptamer and the A124U/A125U double mutant using dimethyl sulfate (DMS), 1-cyclohexyl-[2-morpholinoethyl] carbodiimide metho-p-toluene sulfonate (CMCT), and 1-methyl-7-nitroisatoic anhydride (1M7, SHAPE reagent), as described previously (*Kladwang et al., 2011a*; *Cordero et al., 2012*; *Kladwang et al., 2014*). Briefly, RNAs were transcribed as above with unmodified NTPs only, then PAGE-purified as above on a denaturing 8% polyacrylamide/7 M urea gel, using UV-shadowing of a duplicate sample in an adjacent lane to avoid UV-induced damage (*Kladwang et al., 2012b*). For 1D chemical mapping, 1.2 pmol of RNA was folded with AdoCbl as described above in 15 µl samples, then modified by adding 5 µl of freshly prepared modification reagent (0.5% DMS, 10.5 mg/ml CMCT, or 4.2 mg/ml 1M7) and incubating at room temperature for 10 min (1M7) or 15 min (DMS, CMCT). RNase-free $H_2O$ was added to a no-modification control reaction. Reactions were quenched and RNA was purified and reverse-transcribed as described previously (*Tian et al., 2014*), then cDNAs were purified, eluted from magnetic beads, and analyzed on an ABI3130xl capillary electrophoresis instrument

(Applied Biosystems) (*Tian et al., 2014*). Sequencing ladders were also prepared in parallel (*Tian et al., 2014*). The HiTRACE software package version 2.0 (MATLAB toolbox available at https://github.com/hitrace [*Yoon et al., 2011*] and web server available at http://hitrace.org [*Kim et al., 2013*]) was used to analyze CE data as described previously, including normalization against GAGUA reference hairpins included at the 5′- and 3′-ends of each construct (*Kladwang et al., 2014*; *Tian et al., 2014*).

## Computational modeling

### De novo RNA modeling with $M^2$ and MOHCA-seq constraints

We used the Rosetta software (version r56277) to model all RNAs of interest in three steps (software available at https://www.rosettacommons.org/software) (*Cheng et al., 2015*). Modeling for the AdoCbl riboswitch aptamer was performed with some differences, as noted below. First, we pre-assembled the helix regions of the RNAs using FARNA with full-atom refinement (FARFAR), using the python script helix_preassemble_setup.py. Pre-assembling helices reduced overall computational expense by removing sampling of secondary structure elements. For each helical region, 100 FARFAR models were generated, with the following sample Rosetta command line:

    rna_denovo -nstruct 100 -params_file helix.params -fasta helix.fasta -out:file:silent helix0.out
    -include_neighbor_base_stacks -minimize_rna true -rna::corrected_geo -score:rna_torsion_poten-
    tial RNA11_based_new -geom_sol_correct_acceptor_base -chemical::enlarge_H_lj -score:weights
    rna/rna_helix -cycles 1000 -output_res_num 136-142 221-227

For small RNAs, such as the c-di-GMP riboswitch aptamer, we allowed conformational sampling of helix elements by omitting helix pre-assembly and instead including pseudo-energy constraints (as described below) for secondary structure elements.

Second, we performed low-resolution modeling using the pre-assembled helices and tertiary constraints derived from maxima in the 2D MOHCA-seq proximity map. Tertiary constraints were determined by identifying peaks from analyzed data that were (1) distinguishable from the local background signal by unbiased inspection and (2) not attributable to secondary structure, based on secondary structures inferred by $M^2$ (*Kladwang et al., 2011a*). This produced a list of pairs of residues that were suggested to be spatially proximal by the MOHCA-seq data (tabulated in *Supplementary file 2*). We sorted the selected pairs for each RNA into strong and weak hits based on the apparent intensity of the signal relative to the background. For each pair of residues showing a strong MOHCA-seq hit, we constrained the distance between the O2′ of first residue and C4′ of the second residue using a potential of the following form (see graph in *Figure 1—figure supplement 2A*):

$$S(x) = 3x^2 - 2x^3,$$

$$E(d) = \begin{cases} 4(1 - S(d/15)) & \text{if } 0 \leq d < 15 \\ 4S(d/15 - 1) & \text{if } 15 \leq d < 30 \\ 4 + 36S(d/30 - 1) & \text{if } 30 \leq d < 60 \\ 40 & \text{if } d \geq 60 \end{cases}.$$

Here, $S$ is the smoothstep function, $d$ is the distance between the atom pair, and $E$ is the constraint potential in Rosetta energy units. For residue pairs with a weak MOHCA-seq signal, we applied a constraint potential of the same shape but weaker amplitude (1/5 of the original potential).

With the constraints and the pre-assembled helices, we performed an FARNA simulation to generate a large number of low-resolution Rosetta models (ranging from 10,000 to 60,000). For cases in which other experimental constraints (hydroxyl radical footprinting for the P4–P6 domain and class I ligase; tertiary information from $M^2$ on the P4–P6 domain) were available, these constraints were also included in the modeling. To assess the importance of MOHCA-seq constraints for accurate modeling, we also performed modeling without MOHCA-seq constraints ($M^2$/Rosetta modeling) using the same command lines as above but without the flags to input constraint files and stage

constraints (–cst_file constraint and -staged_constraints, below). A sample Rosetta command line is shown below:

```
rna_denovo -nstruct 500 -params_file rna_params.params -fasta rna_fasta.fasta -out:file:silent rna.
out -include_neighbor_base_stacks -minimize_rna false -native rna_native.pdb -in:file:silent helix0.
out helix1.out helix2.out helix3.out helix4.out helix5.out helix6.out -input_res 1-8 19-26 12-17 122-
127 42-46 84-88 55-61 113-119 62-66 72-76 79-82 89-94 111-112 –cst_file constraint -stage-
d_constraints -cycles 20000 -output_res_num 1-127
```

In the final step, we refined models from the initial run that had low Rosetta low-resolution scores (within the lowest 1/6 of the models) using the high-resolution Rosetta score to obtain final minimized models. This was achieved using the rna_minimize Rosetta application:

```
rna_minimize  -native  rna_native.pdb  -cst_fa_file  constraint  -params_file  rna_params.params
-skip_coord_constraints -in:file:silent 0.silent -out:file:silent rna_min.out
```

To find representative models, we clustered the lowest energy models (3% of the total refined models or 0.5% of the total number of the unrefined models), with a threshold (based on all-heavy-atom RMSD) chosen so as to give 1/6 of the clustered models in the most populous cluster, as in prior work (**Das et al., 2008**). The clustering was achieved by first finding the model (the 'cluster center') with the largest number of neighboring models within the RMSD threshold, then assigning these models to the first cluster (cluster 0). This process was then repeated to cluster all of the remaining models. The cluster center model represents the best tertiary structure prediction of the MCM pipeline. Clustering was performed using Rosetta, with the following command line:

```
cluster -in:file:silent silent.out -in:file:fullatom -out:file:silent_struct_type binary_rna -export_only_-
low false -out:file:silent cluster.out -cluster:radius 7
```

The RMSD threshold described above represents an estimate of the intra-cluster RMSD of the structure ensemble from the modeling. We used this intra-cluster RMSD to calculate an in situ RMSD estimate ('precision') of the ensemble to the 'true' structure. The in situ RMSD of the cluster center model to the 'true' structure can be stated as the distance in structure space between the 'true' structure and the cluster center. This can be conservatively estimated as the sum in quadrature of the RMSDs between (1) the 'true' structure and the most accurate member of the cluster and (2) this cluster member and the cluster center. The intra-cluster RMSD represents an upper bound on distance (2), and assuming that distance (1) is no greater than distance (2), the in situ RMSD of the cluster center to the 'true' structure is estimated as the intra-cluster RMSD multiplied by the square root of 2. This precision estimate is reported in *Table 2*. The accuracy of the ensemble was estimated using the RMSD of the cluster center of the largest cluster to the gold-standard crystal structure. The accuracy and precision of modeling for the AdoCbl riboswitch aptamer were determined slightly differently, as described below, to allow direct comparison to prior modeling efforts in the RNA-puzzle challenge.

As another measure of prediction confidence, we calculated the p-value of each final MCM and M²/Rosetta model using a previously described empirical formula that relates mean pairwise phosphate backbone RMSD to RNA length to allow for calculation of a prediction significance (**Hajdin et al., 2010**). These values are reported for the benchmark RNAs in *Table 1*. Finally, we calculated the percentage of strong and weak MOHCA-seq constraints satisfied by our models (averaged over all models in each cluster) and the crystal structure. We defined a constraint as 'satisfied' if the O2′ of the first residue was within 30 Å of the C4′ of the second residue. The cluster center models produced by MCM modeling generally satisfied 80% or more of the strong constraints and 60% or more of the weak constraints. As expected, the models generally satisfied a greater percentage of constraints than the crystal structures; however, the percent of constraints satisfied was not correlated with the accuracy or precision of the models or with the level of cleavage observed in the proximity maps. As described in the main text, unsatisfied constraints could arise from structural perturbations due to radical source attachment, or conformational flexibility of the RNA, or the benzyl-thiourea linker tethering the radical source to the RNA backbone. The details of the modeling results and these statistics are listed in *Tables 1, 2*.

For the AdoCbl riboswitch aptamer (RNA-puzzle 6) and the lariat-capping ribozyme (RNA-puzzle 5), we did not pre-assemble helices and instead used prebuilt RNA fragments that we previously

generated during the RNA-puzzles challenges, as described in (*Miao et al., 2015*). The prior models generated for the RNA-puzzles 5–6 challenges can be viewed on the RNA-puzzles website (http://ahsoka.u-strasbg.fr/rnapuzzles/problems_past.html). For the AdoCbl riboswitch aptamer, the pre-built fragments consisted of three models each of the P1 through P6, P7 through P8, and P10 through P11 regions. We modeled the AdoCbl riboswitch aptamer with three distinct combinations of these fragments, each containing one fragment of each region. To estimate the precision of MCM modeling for the AdoCbl riboswitch aptamer in the ligand-bound and ligand-free states, we calculated the average pairwise all-heavy-atom RMSD between the cluster centers of the largest clusters for the three modeling setups and used this as the intra-cluster RMSD for calculating the in situ precision estimate.

For the lariat-capping ribozyme, we used MCM to guide rebuilding of the peripheral regions of the structure, and RMSD accuracies to the crystal structure and constraint satisfaction percentages are reported separately for these peripheral regions and the full structure (*Tables 1, 2*). RMSD accuracies of only the refined regions of the structure were calculated by aligning the models to the crystal structure using MAMMOTH with the distance threshold set to the default, 4 Å, and then computing the RMSD of the refined residues (*Ortiz et al., 2002*). To estimate the precision of the refined regions, we performed the same alignment, comparing each cluster model to the cluster center, and used the highest RMSD as the intra-cluster RMSD. To calculate the percentage of constraints satisfied for the refined regions, we used only constraints that included at least one residue within the peripheral regions, to assess how MCM affected its orientation to both itself and to the rest of the structure. For global constraint satisfaction percentages, we used all constraints.

For the class I ligase, we calculated RMSD accuracies to the crystal structure and constraint satisfaction percentages separately for the ribozyme core (excluding the substrate helix) and the full structure. RMSD accuracies of the core by itself were calculated by aligning the core residues, including the active site (residues 28–114), of the models to the crystal structure using MAMMOTH with the distance threshold set to 20 Å, then computing the RMSD of the refined residues (*Ortiz et al., 2002*). To estimate the precision of the core by itself, we performed the same alignment, comparing each cluster model to the cluster center, and used the highest RMSD as the intra-cluster RMSD. To calculate the percentage of constraints satisfied for the core by itself, we used only constraints for which both residues were in the core. For global constraint satisfaction percentages, we used all constraints.

For the ligand-free states of the c-di-GMP, glycine, and AdoCbl riboswitch aptamers, we performed modeling as above with pseudo-energy constraints from MOHCA-seq proximity maps collected in ligand-free conditions. Because the ligand-free states are expected to exhibit greater conformational heterogeneity, we omitted the clustering step and compared the minimized models with the lowest Rosetta energy scores for visualization. We found that for constructs with weak residual tertiary MOHCA-seq hits, such as the c-di-GMP riboswitch aptamer and glycine riboswitch ligand-binding domain with leader sequence, these lowest energy models included both compact or native-like conformations as well as extended and incorrectly folded conformations. For the AdoCbl riboswitch aptamer, we compared the cluster centers of the three independent modeling setups as a similar way of assessing heterogeneity between modeling runs. These comparisons are illustrated in *Figures 4C*, *5C*, *6B*.

For the HoxA9 IRES domain, we performed modeling with two secondary structures, one including the pseudoknot predicted by M$^2$ and mutate-and-rescue, and one without the pseudoknot, predicted by only M$^2$ data. When the pseudoknot was not specified in the secondary structure, a MOHCA-seq pseudo-energy constraint was included during modeling to reflect the proximity observed by the MOHCA-seq experiment (constraint pair in parentheses, *Supplementary file 2*). Our modeling generated global structures for the domain with a similar core region between the models for the two secondary structures (*Figure 7C–D*). As an additional estimate of the modeling resolution, we calculated the RMSD between the cluster center for the top cluster modeled with and without the pseudoknot, in addition to the standard estimation of the precision for each top cluster, and found that the RMSD between cluster centers (19.5 Å) was similar to the precision within each cluster (19.5 Å and 19.8 Å for with and without pseudoknot, respectively) (*Table 2*).

All-heavy-atom RMSDs for RNA-puzzle models (*Figure 3D,F* and *Figure 3—figure supplement 4*) were calculated using the rna_score application in Rosetta with the following command line:

```
rna_score -s model.pdb -native native.pdb
```

where the native PDB was the crystal structure (*Peselis & Serganov, 2012*; *Trausch et al., 2014*; *Zhang & Ferre-D'Amare, 2013*). For the SAM-I/IV riboswitch aptamer (RNA-puzzle 8), the last six

nucleotides at the 3′-end of the crystal RMSD did not match the RNA-puzzle sequence, so the RMSD was calculated over the regions of matching sequence, residues 1–91. For the T-box/tRNA$^{Gly}$ (RNA-puzzle 10), the first nucleotide and last nine nucleotides of the tRNA in the co–crystal structure did not match the RNA-puzzle sequence, so the RMSD was calculated over the regions of matching sequence, residues 6–66; the RMSD calculations included all T-box residues.

## Acknowledgements

We thank members of the Das lab, M Barna, V Risca, W Greenleaf, and S Xue for discussions and comments on the manuscript.

## Additional information

### Funding

| Funder | Grant reference | Author |
|---|---|---|
| National Institutes of Health (NIH) | 5 T32 GM007276 | Clarence Yu Cheng |
| Burroughs Wellcome Fund | CASI 1007326.01 | Rhiju Das |
| Stanford University | Bio-X Graduate Fellowship | Fang-Chieh Chou |
| Howard Hughes Medical Institute (HHMI) | International Student Research Fellowship | Fang-Chieh Chou |
| Consejo Nacional de Ciencia y Tecnología | CONACyT Fellowship 312765 | Pablo Cordero |
| Stanford University | Stanford Graduate Fellowship in Science and Engineering | Siqi Tian |
| National Institutes of Health (NIH) | R01 GM102519 | Rhiju Das |

The funders had no role in study design, data collection and interpretation, or the decision to submit the work for publication.

### Author contributions

CYC, Conception and design, Acquisition of data, Analysis and interpretation of data, Drafting or revising the article; F-CC, Acquisition of data, Analysis and interpretation of data, Drafting or revising the article; WK, ST, PC, Acquisition of data, Analysis and interpretation of data; RD, Conception and design, Analysis and interpretation of data, Drafting or revising the article

## Additional files

### Supplementary files

• Supplementary file 1. Sequences of RNAs, single-stranded DNA ligation adapters, and sequencing primers.

• Supplementary file 2. Pairwise MOHCA-seq constraints used for MCM modeling.

• Source code 1. Models of HoxA9 5′-UTR 957–1132 IRES domain generated by MCM.

### Major datasets

The following datasets were generated:

| Author(s) | Year | Dataset title | Dataset ID and/or URL | Database, license, and accessibility information |
|---|---|---|---|---|
| Cheng CY, Chou F-C, Kladwang W, Tian S, Cordero P, Das R | 2015 | MOHCA-seq data for Tetrahymena ribozyme P4-P6 domain | http://rmdb.stanford.edu/detail/TRP4P6_MCA_0000 | Publicly available at RNA Mapping Database (RMDB) TRP4P6_MCA_0000. |

| Author(s) | Year | Dataset title | Dataset ID and/or URL | Database, license, and accessibility information |
|---|---|---|---|---|
| Cheng CY, Chou F-C, Kladwang W, Tian S, Cordero P, Das R | 2015 | MOHCA-seq data for RNA-puzzle 6, S. thermophilum adenosylcobalamin riboswitch | http://rmdb.stanford.edu/detail/RNAPZ6_MCA_0000 | Publicly available at RNA Mapping Database (RMDB) RNAPZ6_MCA_0000. |
| Cheng CY, Chou F-C, Kladwang W, Tian S, Cordero P, Das R | 2015 | MOHCA-seq data for RNA-puzzle 5, D. iridis lariat-capping ribozyme | http://rmdb.stanford.edu/detail/RNAPZ5_MCA_0000 | Publicly available at RNA Mapping Database (RMDB) Publicly available at RNA Mapping Database (RMDB) RNAPZ5_MCA_0000. |
| Cheng CY, Chou F-C, Kladwang W, Tian S, Cordero P, Das R | 2015 | MOHCA-seq data for V. cholerae cyclic-di-GMP riboswitch | http://rmdb.stanford.edu/detail/CDIGMP_MCA_0000 | Publicly available at RNA Mapping Database (RMDB) CDIGMP_MCA_0000. |
| Cheng CY, Chou F-C, Kladwang W, Tian S, Cordero P, Das R | 2015 | MOHCA-seq data for F. nucleatum glycine riboswitch | http://rmdb.stanford.edu/detail/GLYCFN_MCA_0000 | Publicly available at RNA Mapping Database (RMDB) in the public domain. |
| Cheng CY, Chou F-C, Kladwang W, Tian S, Cordero P, Das R | 2015 | MOHCA-seq data for F. nucleatum glycine riboswitch | http://rmdb.stanford.edu/detail/GLYCFN_MCA_0001 | Publicly available at RNA Mapping Database (RMDB) in the public domain. |
| Cheng CY, Chou F-C, Kladwang W, Tian S, Cordero P, Das R | 2015 | MOHCA-seq data for F. nucleatum glycine riboswitch | http://rmdb.stanford.edu/detail/GLYCFN_MCA_0002 | Publicly available at RNA Mapping Database (RMDB) in the public domain. |
| Cheng CY, Chou F-C, Kladwang W, Tian S, Cordero P, Das R | 2015 | MOHCA-seq data for F. nucleatum glycine riboswitch | http://rmdb.stanford.edu/detail/GLYCFN_MCA_0003 | Publicly available at RNA Mapping Database (RMDB) in the public domain. |
| Cheng CY, Chou F-C, Kladwang W, Tian S, Cordero P, Das R | 2015 | MOHCA-seq data for F. nucleatum glycine riboswitch with leader sequence | http://rmdb.stanford.edu/detail/GLYCFN_KNK_0003 | Publicly available at RNA Mapping Database (RMDB) in the public domain. |
| Cheng CY, Chou F-C, Kladwang W, Tian S, Cordero P, Das R | 2015 | MOHCA-seq data for F. nucleatum glycine riboswitch with leader sequence | http://rmdb.stanford.edu/detail/GLYCFN_KNK_0004 | Publicly available at RNA Mapping Database (RMDB) in the public domain. |
| Cheng CY, Chou F-C, Kladwang W, Tian S, Cordero P, Das R | 2015 | MOHCA-seq data for F. nucleatum glycine riboswitch with leader sequence | http://rmdb.stanford.edu/detail/GLYCFN_KNK_0005 | Publicly available at RNA Mapping Database (RMDB) in the public domain. |
| Cheng CY, Chou F-C, Kladwang W, Tian S, Cordero P, Das R | 2015 | MOHCA-seq data for F. nucleatum glycine riboswitch with leader sequence | http://rmdb.stanford.edu/detail/GLYCFN_KNK_0006 | Publicly available at RNA Mapping Database (RMDB) in the public domain. |
| Cheng CY, Chou F-C, Kladwang W, Tian S, Cordero P, Das R | 2015 | MOHCA-seq data for F. nucleatum glycine riboswitch with leader sequence | http://rmdb.stanford.edu/detail/GLYCFN_KNK_0007 | Publicly available at RNA Mapping Database (RMDB) in the public domain. |

| Author(s) | Year | Dataset title | Dataset ID and/or URL | Database, license, and accessibility information |
|---|---|---|---|---|
| Cheng CY, Chou F-C, Kladwang W, Tian S, Cordero P, Das R | 2015 | MOHCA-seq data for F. nucleatum glycine riboswitch with leader sequence | http://rmdb.stanford.edu/detail/GLYCFN_KNK_0008 | Publicly available at RNA Mapping Database (RMDB) in the public domain. |
| Cheng CY, Chou F-C, Kladwang W, Tian S, Cordero P, Das R | 2015 | MOHCA-seq data for F. nucleatum glycine riboswitch with leader sequence | http://rmdb.stanford.edu/detail/GLYCFN_KNK_0009 | Publicly available at RNA Mapping Database (RMDB) in the public domain. |
| Cheng CY, Chou F-C, Kladwang W, Tian S, Cordero P, Das R | 2015 | MOHCA-seq data for F. nucleatum glycine riboswitch with leader sequence | http://rmdb.stanford.edu/detail/GLYCFN_KNK_0010 | Publicly available at RNA Mapping Database (RMDB) in the public domain. |
| Cheng CY, Chou F-C, Kladwang W, Tian S, Cordero P, Das R | 2015 | MOHCA-seq data for class I ligase | http://rmdb.stanford.edu/detail/CL1LIG_MCA_0000 | Publicly available at RNA Mapping Database (RMDB) in the public domain. |
| Cheng CY, Chou F-C, Kladwang W, Tian S, Cordero P, Das R | 2015 | MOHCA-seq data for class I ligase | http://rmdb.stanford.edu/detail/CL1LIG_MCA_0001 | Publicly available at RNA Mapping Database (RMDB) in the public domain. |
| Cheng CY, Chou F-C, Kladwang W, Tian S, Cordero P, Das R | 2015 | MOHCA-seq data for class I ligase | http://rmdb.stanford.edu/detail/CL1LIG_MCA_0002 | Publicly available at RNA Mapping Database (RMDB) in the public domain. |
| Cheng CY, Chou F-C, Kladwang W, Tian S, Cordero P, Das R | 2015 | MOHCA-seq data for class I ligase | http://rmdb.stanford.edu/detail/CL1LIG_MCA_0003 | Publicly available at RNA Mapping Database (RMDB) in the public domain. |
| Cheng CY, Chou F-C, Kladwang W, Tian S, Cordero P, Das R | 2015 | MOHCA-seq data for HoxA9 5'-UTR 957-1132 IRES domain | http://rmdb.stanford.edu/detail/HOXA9D_MCA_0000 | Publicly available at RNA Mapping Database (RMDB) in the public domain. |
| Cheng CY, Chou F-C, Kladwang W, Tian S, Cordero P, Das R | 2015 | MOHCA-seq data for HoxA9 5'-UTR 957-1132 IRES domain | http://rmdb.stanford.edu/detail/HOXA9D_MCA_0001 | Publicly available at RNA Mapping Database (RMDB) in the public domain. |
| Cheng CY, Chou F-C, Kladwang W, Tian S, Cordero P, Das R | 2015 | MOHCA-seq data for Tetrahymena ribozyme P4-P6 domain | http://rmdb.stanford.edu/detail/TRP4P6_MCA_0001 | Publicly available at RNA Mapping Database (RMDB) TRP4P6_MCA_0001. |
| Cheng CY, Chou F-C, Kladwang W, Tian S, Cordero P, Das R | 2015 | MOHCA-seq data for Tetrahymena ribozyme P4-P6 domain | http://rmdb.stanford.edu/detail/TRP4P6_MCA_0002 | Publicly available at RNA Mapping Database (RMDB) TRP4P6_MCA_0002. |
| Cheng CY, Chou F-C, Kladwang W, Tian S, Cordero P, Das R | 2015 | MOHCA-seq data for Tetrahymena ribozyme P4-P6 domain | http://rmdb.stanford.edu/detail/TRP4P6_MCA_0003 | Publicly available at RNA Mapping Database (RMDB) TRP4P6_MCA_0003. |
| Cheng CY, Chou F-C, Kladwang W, Tian S, Cordero P, Das R | 2015 | MOHCA-seq data for Tetrahymena ribozyme P4-P6 domain | http://rmdb.stanford.edu/detail/TRP4P6_MCA_0004 | Publicly available at RNA Mapping Database (RMDB) TRP4P6_MCA_0004. |

| Author(s) | Year | Dataset title | Dataset ID and/or URL | Database, license, and accessibility information |
|---|---|---|---|---|
| Cheng CY, Chou F-C, Kladwang W, Tian S, Cordero P, Das R | 2015 | MOHCA-seq data for RNA-puzzle 6, S. thermophilum adenosylcobalamin riboswitch | http://rmdb.stanford.edu/detail/RNAPZ6_MCA_0001 | Publicly available at RNA Mapping Database (RMDB) RNAPZ6_MCA_0001. |
| Cheng CY, Chou F-C, Kladwang W, Tian S, Cordero P, Das R | 2015 | MOHCA-seq data for RNA-puzzle 6, S. thermophilum adenosylcobalamin riboswitch | http://rmdb.stanford.edu/detail/RNAPZ6_MCA_0002 | Publicly available at RNA Mapping Database (RMDB) RNAPZ6_MCA_0002. |
| Cheng CY, Chou F-C, Kladwang W, Tian S, Cordero P, Das R | 2015 | MOHCA-seq data for RNA-puzzle 6, S. thermophilum adenosylcobalamin riboswitch | http://rmdb.stanford.edu/detail/RNAPZ6_MCA_0003 | Publicly available at RNA Mapping Database (RMDB) RNAPZ6_MCA_0003. |
| Cheng CY, Chou F-C, Kladwang W, Tian S, Cordero P, Das R | 2015 | MOHCA-seq data for RNA-puzzle 6, S. thermophilum adenosylcobalamin riboswitch | http://rmdb.stanford.edu/detail/RNAPZ6_MCA_0004 | Publicly available at RNA Mapping Database (RMDB) RNAPZ6_MCA_0004. |
| Cheng CY, Chou F-C, Kladwang W, Tian S, Cordero P, Das R | 2015 | MOHCA-seq data for RNA-puzzle 6, S. thermophilum adenosylcobalamin riboswitch | http://rmdb.stanford.edu/detail/RNAPZ6_MCA_0005 | Publicly available at RNA Mapping Database (RMDB) RNAPZ6_MCA_0005. |
| Cheng CY, Chou F-C, Kladwang W, Tian S, Cordero P, Das R | 2015 | MOHCA-seq data for RNA-puzzle 6, S. thermophilum adenosylcobalamin riboswitch | http://rmdb.stanford.edu/detail/RNAPZ6_MCA_0006 | Publicly available at RNA Mapping Database (RMDB) RNAPZ6_MCA_0006. |
| Cheng CY, Chou F-C, Kladwang W, Tian S, Cordero P, Das R | 2015 | MOHCA-seq data for RNA-puzzle 6, S. thermophilum adenosylcobalamin riboswitch | http://rmdb.stanford.edu/detail/RNAPZ6_MCA_0007 | Publicly available at RNA Mapping Database (RMDB) RNAPZ6_MCA_0007. |
| Cheng CY, Chou F-C, Kladwang W, Tian S, Cordero P, Das R | 2015 | MOHCA-seq data for RNA-puzzle 5, D. iridis lariat-capping ribozyme | http://rmdb.stanford.edu/detail/RNAPZ5_MCA_0001 | Publicly available at RNA Mapping Database (RMDB) Publicly available at RNA Mapping Database (RMDB) RNAPZ5_MCA_0001. |
| Cheng CY, Chou F-C, Kladwang W, Tian S, Cordero P, Das R | 2015 | MOHCA-seq data for RNA-puzzle 5, D. iridis lariat-capping ribozyme | http://rmdb.stanford.edu/detail/RNAPZ5_MCA_0002 | Publicly available at RNA Mapping Database (RMDB) RNAPZ5_MCA_0002. |
| Cheng CY, Chou F-C, Kladwang W, Tian S, Cordero P, Das R | 2015 | MOHCA-seq data for V. cholerae cyclic-di-GMP riboswitch | http://rmdb.stanford.edu/detail/CDIGMP_MCA_0001 | Publicly available at RNA Mapping Database (RMDB) CDIGMP_MCA_0001. |
| Cheng CY, Chou F-C, Kladwang W, Tian S, Cordero P, Das R | 2015 | MOHCA-seq data for V. cholerae cyclic-di-GMP riboswitch | http://rmdb.stanford.edu/detail/CDIGMP_MCA_0002 | Publicly available at RNA Mapping Database (RMDB) CDIGMP_MCA_0002. |
| Cheng CY, Chou F-C, Kladwang W, Tian S, Cordero P, Das R | 2015 | MOHCA-seq data for V. cholerae cyclic-di-GMP riboswitch | http://rmdb.stanford.edu/detail/CDIGMP_MCA_0003 | Publicly available at RNA Mapping Database (RMDB) CDIGMP_MCA_0003. |

| Author(s) | Year | Dataset title | Dataset ID and/or URL | Database, license, and accessibility information |
|---|---|---|---|---|
| Cheng CY, Chou F-C, Kladwang W, Tian S, Cordero P, Das R | 2015 | MOHCA-seq data for V. cholerae cyclic-di-GMP riboswitch | http://rmdb.stanford.edu/detail/CDIGMP_MCA_0004 | Publicly available at RNA Mapping Database (RMDB) CDIGMP_MCA_0004. |
| Cheng CY, Chou F-C, Kladwang W, Tian S, Cordero P, Das R | 2015 | MOHCA-seq data for V. cholerae cyclic-di-GMP riboswitch | http://rmdb.stanford.edu/detail/CDIGMP_MCA_0005 | Publicly available at RNA Mapping Database (RMDB) CDIGMP_MCA_0005. |
| Cheng CY, Chou F-C, Kladwang W, Tian S, Cordero P, Das R | 2015 | MOHCA-seq data for V. cholerae cyclic-di-GMP riboswitch | http://rmdb.stanford.edu/detail/CDIGMP_MCA_0006 | Publicly available at RNA Mapping Database (RMDB) CDIGMP_MCA_0006. |
| Cheng CY, Chou F-C, Kladwang W, Tian S, Cordero P, Das R | 2015 | MOHCA-seq data for V. cholerae cyclic-di-GMP riboswitch | http://rmdb.stanford.edu/detail/CDIGMP_MCA_0007 | Publicly available at RNA Mapping Database (RMDB) CDIGMP_MCA_0007. |

The following previously published datasets were used:

| Author(s) | Year | Dataset title | Dataset ID and/or URL | Database, license, and accessibility information |
|---|---|---|---|---|
| Cate JH, Gooding AR, Podell E, Zhou K, Golden BL, Kundrot CE, Cech TR, Doudna JA | 1996 | Crystal structure of a group I ribozyme domain: principles of RNA packing | http://www.rcsb.org/pdb/explore/explore.do?structureId=1GID | Publicly available at RCSB Protein Data Bank (Accession No: 1GID). |
| Smith KD, Lipchock SV, Ames TD, Wang J, Breaker RR, Strobel SA | 2009 | Structure of a c-di-GMP riboswitch from V. cholerae | http://www.rcsb.org/pdb/explore/explore.do?structureId=3IRW | Publicly available at RCSB Protein Data Bank (Accession No: 3IRW). |
| Butler EB, Xiong Y, Wang J, Strobel SA | 2011 | Crystal Structure of a Glycine Riboswitch from Fusobacterium nucleatum | http://www.rcsb.org/pdb/explore/explore.do?structureId=3P49 | Publicly available at RCSB Protein Data Bank (Accession No: 3P49). |
| Peselis A, Serganov A | 2012 | RNA structure | http://www.rcsb.org/pdb/explore/explore.do?structureId=4GXY | Publicly available at RCSB Protein Data Bank (Accession No: 4GXY). |
| Shechner DM, Grant RA, Bagby SC, Koldobskaya Y, Piccirilli JA, Bartel DP | 2009 | Crystal structure of class I ligase ribozyme self-ligation product, in complex with U1A RBD | http://www.rcsb.org/pdb/explore/explore.do?structureId=3HHN | Publicly available at RCSB Protein Data Bank (Accession No: 3HHN). |
| Meyer M, Nielsen H, Olieric V, Roblin P, Johansen SD, Westhof E, Masquida B | 2014 | Speciation of a group I intron into a lariat capping ribozyme | http://www.rcsb.org/pdb/explore/explore.do?structureId=4P8Z | Publicly available at RCSB Protein Data Bank (Accession No: 4P8Z). |
| Blaha G, Gurel G, Schroeder SJ, Moore PB, Steitz TA | 2008 | The Refined Crystal Structure of the Haloarcula Marismortui Large Ribosomal Subunit at 2.4 Angstrom Resolution with rrnA Sequence for the 23S rRNA and Genome-derived Sequences for r-Proteins | http://www.rcsb.org/pdb/explore/explore.do?structureId=3CC2 | Publicly available at RCSB Protein Data Bank (Accession No: 3CC2). |

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
