## [Decision Letter]

Thank you for resubmitting your work entitled “Consistent global structures of complex RNA states through multidimensional chemical mapping” for further consideration at *eLife*. Your revised article has been favorably evaluated by James Manley (Senior editor), the member of the Board of Reviewing Editors who oversaw the original review, and the three original peer reviewers. As you will see, all thought that the paper was significantly improved; thank you for the very conscientious revision. Please address the remaining concerns raised by referee 1 before uploading your final files.

*Reviewer #1*:

The revised manuscript by Cheng et al. on modeling of RNA structures is greatly improved over the original submission. In particular, it is now possible to understand the method from the main text, and the controls and assumptions are explained more clearly. These explanations also better demonstrate the usefulness and potential of this method for probing and modeling RNA structures. Overall, this is very interesting and significant work.

I still have a few questions that were not fully addressed by the authors in their revisions.

Specific comments:

Figure 3: The authors now explain the effects of oxidative damage far more clearly. Nevertheless, the very high frequency of additional RT stops could have a negative effect on the modeling, which the authors should discuss. First, by generating reads that are shorter than those arising from RT stops at site of the tether, they inflate the number of short pairwise constraints. It is still not clear to me that these do not bias the modeling procedure. Second, some of these reads could arise from two damaged sites 5' of the tether, producing false constraints.

In Figure 3: Previous authors (e.g. Brenowitz et al.) reported a similar frequency of RT stops and strand cleavage, and not the 6-fold difference reported here. Could the authors comment on the reason for this difference? Is this because the authors use only ascorbate, rather than a combination of hydrogen peroxide and ascorbate? I am still puzzled by the claim that these additional RT stops arise from common base oxidation products, such as 8-oxoG or 5'-HOC. Reverse transcriptases (and other DNA polymerases) are reported in the literature to bypass such lesions fairly easily.

In response to the reviewer comments, the authors provide better comparisons of models with and without the MOHCA constraints, but they didn't address how sensitive the models are to the number (and quality) of the constraints. If you leave out the “false” constraints, are the models better? If you include only half the number of “true” constraints, are the models significantly worse? This is certainly worth discussing and testing (if not now, in the future).

The comparison of liganded and free riboswitches in Figures 6 and 7 are a nice demonstration that the method is robust enough to give some kind of prediction even when the RNA is not stably folded. The conclusion that the models represent the actual ensemble of unliganded RNA conformations, however, is untested, and probably incorrect. The authors should remove this claim from the discussion and figure legends. It is easy to see from Figures 6 and 7 that the uncertainty in modeling the unliganded riboswitches comes from a lack of constraints, rather than an increase in the average distance between RNA helices or a greater variety of constraints. Moreover, they necessarily obtain native-like structures in the absence of ligand, because these are the only structures that are sufficiently coherent to produce self-consistent constraints. (These two pitfalls are analogous to the well-understood limitations of 3D models based on NMR NOEs.)

[Editors’ note: a previous version of this study was rejected after peer review, but the authors submitted for reconsideration. The previous decision letter after peer review is shown below.]

Thank you for choosing to send your work entitled “Consistent and blind inference of RNA tertiary folds through multidimensional chemical mapping” for consideration at *eLife*. Your full submission has been evaluated by James Manley (Senior editor), a Reviewing editor, and three peer reviewers, and the decision was reached after discussions between the reviewers. Based on our discussions and the individual reviews below, we regret to inform you that your work will not be considered further for publication in *eLife*.

The following individuals responsible for the peer review of your submission have agreed to reveal their identity: Timothy Nilsen (Reviewing editor); Hashim Al-Hashim (reviewer). Two reviewers remain anonymous.

We have received comments on your manuscript from three experts in the specific area of RNA structure analyses. As you will see one reviewer was enthusiastic about the work and recommended publication; however, the other two reviewers, while agreeing that the work was potentially important, raise significant concerns regarding the method you describe. These concerns include technical points as well as uncertainty regarding validation of the method. Given the substantive nature of the issues raised, we must decline your manuscript.

*Reviewer #1*:

The authors describe an improved method for using high throughput sequencing and tethered hydroxyl radical cleavage to obtain distance constraints for RNA modeling. They offer an impressive array of successful examples, and also describe how the method is designed to work in tandem with their previously described mutate and map strategy for obtaining secondary structures. A large amount of work is presented in this paper. Although the pipeline is complicated and beyond the reach of most labs, it would represent a significant step forward, if it proves reliable over the long term. Below I outline some concerns about the presentation of the work and the basic premise of MOHCA-Seq.

The body of the manuscript is more of an advertisement than a description of what was done. The substance of the paper is in the Methods section, and the manuscript should be reorganized so that much more of the modeling strategy and data analysis is brought forward into the “Results” section. Figure 1 and its supplements could be (ought to be) a substantial fraction of the “main” manuscript—this assessment is both a testament to the amount of work described in this paper, and a problem in that critical tests and examples of the method are buried.

Although the authors present many success stories, they do not describe the statistics on how many models are correct, how dependent the final models are on individual constraints (what happens when one leaves some out?), and so on. To help the reader judge the importance of the 3D MOHCA constraints, comparisons in the supplemental Tables 3 and 4 should be brought into the main text and possibly illustrated. A better discussion of the effects of noise, false distance constraints that arise from background cleavage of the RNA or other artifacts is needed.

My main concerns have to do with the distribution of the MOHCA-Seq reads and their interpretation as “distance constraints”. In the main text (and supplement), the authors note that they observe additional sequence reads from arrest of RT at damaged (oxidized) nucleotides that lie between the radical source and the site of cleavage. First, the authors assume these “extra” stops arise from non-scissile base damage, but of course, they could arise from strand scission events. The authors write as if these are in different categories, but this distinction is meaningless as the reason for the RT stop can't be ascertained after the fact.

In [Disp-formula equ4], are the authors simply attempting to model the distribution of their data as arising from a component that depends on the frequency of tethered Fe(II)-EDTA complexes and a component that is independent of having tethered complexes? I presume this is the idea behind the COHCOA correlation analysis, which rightly ignores the distinction between “oxidative damage” and “cleavage”. This could be explained more forthrightly.

On the other hand, as the total dose of Fe(II)EDTA-dependent hydroxyl radical cleavage increases, the frequency of stops at a cleaved residue will equal or exceed RT stops at the site of the tether. In this limit, any information about the pairwise correlation between the site of the tethered reagent and the site of cleavage will be lost, undermining the premise of the experiment. At what limit does this occur, and does this tendency already account for the very uniform map before COHCOA analysis?

Even when cleavage events are sparse, I had trouble understanding how the authors turn the distribution of RT stops into reliable distance constraints. Backbone cleavage arising from a tethered source of hydroxyl radical can be interpreted as a (blurred) vector between two residues. By contrast, sequence reads that start and end at two damaged or cleaved sites may arise when both residues are close to the radical source. However, one cannot be sure how close the sites are to each other, especially as one does not know where the Fe-EDTA was tethered. Presumably the inconsistencies arising from such ambiguities in the distance constraints are sorted out during the modeling, but what evidence do the authors have that this is the case? In the modeling section, it is stated that 60%-80% of constraints were satisfied by the best models; to what degree does this reflect the level of cleavage or specific types of non-random noise (see below)?

[Disp-formula equ3] assumes a uniform rate of background cleavage *b*_*j*_. Of course, the background cleavage is almost certainly non-uniform. It is well known that certain residues will be more prone to hydrolysis (or RNase digestion) than others, and the resulting distribution of reads will appear similar to a genuine signal from a tethered Fe(II)EDTA. Is this type of non-random background addressed by using a mock-treated control? How is the information from the control incorporated into the data analysis? It would be great to see examples of data sets in which the iron was left out, or data sets on a long double-stranded RNA, and so on, to get an idea of how well the method works in these idealized or control examples.

Finally, a drawback of the original MOHCA was that the experimentally observed constraints were all short range with respect to the RNA sequence. A cutoff was applied so that these short-range constraints did not overwhelm the model building. A similar strategy is presumably used here: how sensitive are the models to this cutoff, and what is the practical limit on the size of the RNA that can be modeled by this method?

Reviewer #2:

This manuscript describes important advances in high throughout structure modeling of RNA. The authors show that integrating distance contact information that can readily be obtained from multiplexed hydroxyl radical cleavage analysis with powerful methods developed by the group (mutate-and-map secondary structure analysis and Rosetta modeling), the tertiary fold of large and complex RNAs can be determined with an estimated accuracy of 1 nm. The authors demonstrate their approach on a variety of RNAs including a blind RNA-puzzle. The results are convincing and suggest that the improvements achieved by this new method are both significant and robust.

*Reviewer #3*:

The determination of RNA structures has been challenging. While RNA molecules have evaded easy crystallization at a larger scale for a long time due to their highly charged backbone, structure predictions based sequence has gained large interest. Chemical probing of their structures has complemented these approaches. RNA molecules are comparably simple at the primary sequence and easily accessible to enzymatic and chemical manipulation followed by sequencing.

Cheng et al. present an experimental-computational multi-step pipeline that combines a mutate-and-map approach (M2) with a deep-sequencing based version of multiplexed hydroxyl radical cleavage analysis. The authors use this updated version in what they call a multidimensional chemical mapping (MCM) pipeline to determine a number of RNA structures. Some were known structures in the context of the RNA-puzzle challenge, in part done blindly. Using their pipeline Cheng et al. propose the structures of three riboswitches with and without their ligands and find that those two states are structurally much more similar than previously thought. Attempts to crystallize those switches without ligands may force the molecules into an undesired stable conformation and be non-physiological or these conformations might never result in well diffracting crystals. Lastly they propose a first structure for the human HoxA9D IRES, a cellular IRES implicated in the translational regulation of developmental patterning.

The authors describe an interesting experimental pipeline that may move the determination of RNA structures out of the hands of few specialized RNA biophysics labs into the hands of molecular biologists with access to deep sequencing equipment. The experimental method is relatively straightforward for anybody who has generated deep-sequencing libraries. Even though far from trivial, the adaptation of the protocol can provide a handle to address structural features at a larger scale which is very desirable in different fields of RNA biology.

The method is a further step in structure determination, but has only been verified on a very limited set of RNA structures. This makes some of the predictions difficult to assess. As an example: Figure 2 shows the refinement of P2.1 towards P4-P6. Other structural elements are not refined at all (purple loop). Why is that? Is a global RMSD score the best measure to describe the refinement of the structure? A wrong prediction of functionally critical residues may hamper the use of the method significantly. Can the authors use their data and compare whether the structural refinements are meaningful by focusing on the critical elements for their direct comparison? Due to the linearity of their sequencing approach the method can only detect cleavage that occurs 3' of the •OH source. The authors themselves suggest, that circularization of the RNA might circumvent this issue but argue against it. This assumption may have major consequences on the quality of the prediction.

The manuscript is to a large extent the description of a new method. It is absolutely essential that the protocol is immediately applicable to different questions by different labs. In their Material and methods the authors are not consistent with the information they provide. It would be important if they provide information on how much material they recover at each step of the protocol (Post OH step, 3'end repair, ligation…) as well as the reaction volumes of all steps (Results).

For the description of the riboswitch structures, I prefer to see the apo and the bound structure overlaid to form a better opinion on the extent of structural rearrangement of the two.

I do not feel competent to judge the computational assumptions that have been made by the authors to refine and translate the experimental results to steer the computational structure prediction (Methods).

---

## [Author Response]

*Specific comments*:

Figure 3*: The authors now explain the effects of oxidative damage far more clearly. Nevertheless, the very high frequency of additional RT stops could have a negative effect on the modeling, which the authors should discuss. First, by generating reads that are shorter than those arising from RT stops at site of the tether, they inflate the number of short pairwise constraints. It is still not clear to me that these do not bias the modeling procedure*.

The reviewer is correct that the high frequency of RT stops may cause attenuation of signal for pairs of residues that are far apart in the RNA sequence. Our analysis accounts for this type of attenuation, as described in the Results and Methods sections. We also present in the Methods section an experimental method to enrich long cDNA products and to rigorously combine sequencing data for these samples with non-enriched sequencing data (in the subsection headed ‘Reverse transcription with sequencing primers (first adapter)’).

For the lengths of RNAs probed in our study (up to 188 nucleotides in the region of interest), we were able to observe long-range hits throughout the proximity map, suggesting that our protocol was sufficient to correct for RT attenuation (see MOHCA-seq proximity maps, e.g. Figure 3 and Figure 3—figure supplement 3). For longer RNAs that may be probed using MOHCA-seq in the future, RT attenuation may present a challenge for observing long-range proximities, in which case experimental adjustments to allow read-through of reverse transcription stopping events may need to be leveraged. We have added this point to the Discussion section:

“The experimental cost of MCM grows quadratically with the length of the transcript, limiting current analysis to domains of hundreds of nucleotides; however, >1000-nt transcripts may become feasible as sequencing costs continue to decrease and if new strategies to bypass reverse transcription stops are implemented.”

*Second, some of these reads could arise from two damaged sites 5' of the tether, producing false constraints*.

We realize that our wording was unclear in describing these additional reads that may be due to reverse transcription termination at oxidatively damaged sites. The situation described by the reviewer, in which a strand scission event occurs 5´ of the tethered radical source and reverse transcription starting at that position terminates at a second oxidatively damaged site, should still reflect a ‘true’ constraint similar to constraints produced by strand scission and reverse transcription termination both occurring 3´ of the tethered radical source.

To clarify this point, we have added a panel to Figure 2 (previously Figure 3) with a schematic illustration of the additional RNA fragments that are observable with MOHCA-seq compared to MOHCA-gel, by using 3´-end ligation to the RNA fragments and reverse transcription. We have also added a visualization-only 3D model of the proof-of-concept RNA with these types of hits illustrated on the structure (Figure 2). We have also clarified our wording in the text, specifically in the legend for Figure 2 and in the Results section. In the Results, we now state:

“Since these sites of reverse transcription termination marked locations that were also nearby in three dimensions to the radical source, the 5´ and 3´ ends of the resulting cDNAs would still be expected to report pairs of sites that were proximal in the RNA structure, albeit at longer average distances than source-to-cleavage pairs (illustrated in Figure 2).”

*In*
Figure 3*: Previous authors (e.g. Brenowitz et al.) reported a similar frequency of RT stops and strand cleavage, and not the 6-fold difference reported here. Could the authors comment on the reason for this difference? Is this because the authors use only ascorbate, rather than a combination of hydrogen peroxide and ascorbate? I am still puzzled by the claim that these additional RT stops arise from common base oxidation products, such as 8-oxoG or 5'-HOC. Reverse transcriptases (and other DNA polymerases) are reported in the literature to bypass such lesions fairly easily*.

The exact oxidative lesions present in MOHCA-seq samples that result in RT stops are uncertain. Prior studies indicate that many oxidative modifications to RNA exist that have not been specifically demonstrated to permit or terminate reverse transcription, and the distribution of these modifications between ascorbate- and peroxide/ascorbate-mediated Fenton reactions has also not been characterized. Further experiments may help to identify the particular oxidative lesions that result in RT stops in MOHCA-seq experiments. We have edited the text regarding the reverse transcription-terminating modifications to indicate this (Results section):

“However, we were surprised to find additional stop sites beyond these source attachment positions (Figure 2, right) […] a model hairpin RNA with a single radical source attachment position (Figure 2).”

*In response to the reviewer comments, the authors provide better comparisons of models with and without the MOHCA constraints, but they didn't address how sensitive the models are to the number (and quality) of the constraints. If you leave out the* “*false*” *constraints, are the models better? If you include only half the number of* “*true*” *constraints, are the models significantly worse? This is certainly worth discussing and testing (if not now, in the future)*.

We agree with the reviewer that performing cross-validation, e.g. by leaving out subsets of constraints and testing whether these constraints are recovered by modeling, will be an important next step in improving the method. Such tests would include assessing the effects of ‘true’ or ‘false’ constraints, classified by comparison of the constraints to a crystallographic or other model, on the modeling accuracies. To emphasize this point, we have added text in the Discussion section:

“A fraction of observed MOHCA-seq constraints do not appear to reflect structures solved by crystallography; quantitative relationships correlating MCM data to structural features should further improve modeling accuracy and enable error estimation and cross-validation by leaving out constraints during modeling, analogous to continuing advances in NMR methods.”

*The comparison of liganded and free riboswitches in*
Figures 6 and 7
*are a nice demonstration that the method is robust enough to give some kind of prediction even when the RNA is not stably folded. The conclusion that the models represent the actual ensemble of unliganded RNA conformations, however, is untested, and probably incorrect. The authors should remove this claim from the discussion and figure legends. It is easy to see from*
Figures 6 and 7
*that the uncertainty in modeling the unliganded riboswitches comes from a lack of constraints, rather than an increase in the average distance between RNA helices or a greater variety of constraints. Moreover, they necessarily obtain native-like structures in the absence of ligand, because these are the only structures that are sufficiently coherent to produce self-consistent constraints. (These two pitfalls are analogous to the well-understood limitations of 3D models based on NMR NOEs*.*)*

We thank the reviewer for this comment. To clarify that the models of the unliganded states of the riboswitch aptamers were used for initial visualization only, we have made edits to the Discussion and figure legends, as suggested by the reviewer:

Discussion: “MCM data corroborate and refine models of ligand-independent tertiary structure in a c-di-GMP riboswitch aptamer and reveals similarly preformed tertiary structure in glycine and AdoCbl riboswitch aptamers.”

Figure 4 legend: “(c) Five MCM models with lowest Rosetta energy of c-di-GMP riboswitch aptamer give an initial visualization of ligand-free ensemble (left); modeling included pseudo-energy constraints from 0 µM c-di-GMP and 10 mM Mg^2+^ proximity map.”

Similar edits were made to the legends for Figures 5 and 6.

For additional clarification, we also reiterate in the Results section (subsection headed “Direct nucleotide-resolution observation of preformed riboswitch aptamer tertiary structure using MCM”) that the model ensembles shown for the unliganded riboswitch aptamers were used for initial visualization only:

“As in NMR studies of dynamic molecules, we therefore inspected an ensemble of models with the lowest Rosetta energy score instead of focusing on the cluster center*;* the resulting ensemble was used for initial visualization only. (Salmon et al., 2014, Bothe et al., 2011, Al-Hashimi, 2007)”

“As above, Rosetta modeling produced an initial visualization of a loose structural ensemble that samples, but does not stably remain in, the ligand-bound structure within the nanometer resolution of the MOHCA-seq method (Figure 6 and Table 2).”

[Editors’ note: the author responses to the previous round of peer review follow.]

This is a major reworking of an article previously submitted to *eLife*. We received strong positive comments (‘significant step forward’, ‘important advances…the results are convincing’, ‘very desirable in different fields of RNA biology’) and detailed, valuable suggestions from three reviewers.

The reviewers’ comments explicitly suggested moving several supporting sections with ‘the substance of the paper’ to the main text, which we have done. This has notably improved the clarity of the manuscript. Second, comments asking for more methodological details made us realize that we should have included several control experiments in the paper, especially on detailed quantitation of chemical modification events and on the accuracies of the 3D models without data. In fact, we had these measurements and calculations in hand but had previously left them out due to length concerns; they are now in the revised paper as main text figures. Third, comments from reviewers 2 and 3 on not feeling ‘competent to judge the computational assumptions’ made us realize that the original manuscript did not appropriately define our work’s place in the lineage of hybrid computational/experimental approaches for RNA structure. We have now made major changes to the Title, Abstract, and Introduction to set our work in the context of classic and ongoing research in modelling molecules such as the group I ribozyme, for which global structures at 1-nm resolution have been powerful for fueling new experiments but not consistently attained for general RNAs. Along these lines, we should have suggested more reviewers with computational expertise.

Reviewer #1:

*The authors describe an improved method for using high throughput sequencing and tethered hydroxyl radical cleavage to obtain distance constraints for RNA modeling. They offer an impressive array of successful examples, and also describe how the method is designed to work in tandem with their previously described mutate and map strategy for obtaining secondary structures. A large amount of work is presented in this paper. Although the pipeline is complicated and beyond the reach of most labs, it would represent a significant step forward, if it proves reliable over the long term*.

We thank the reviewer for their detailed assessment of our work. While we agree that our pipeline involves several steps for library preparation, we believe (as does reviewer #3 below) that the method is not more complicated than RNA-seq and related protocols that are entering wide use, such as ribosome profiling (Ribo-Seq), crosslinking and immunoprecipitation sequencing (CLIP-seq), or crosslinking ligation and sequencing of hybrids (CLASH), and should be accessible to most labs.

*Below I outline some concerns about the presentation of the work and the basic premise of MOHCA-Seq*.

*The body of the manuscript is more of an advertisement than a description of what was done. The substance of the paper is in the Methods section, and the manuscript should be reorganized so that much more of the modeling strategy and data analysis is brought forward into the* “*Results*” *section.*
Figure 1
*and its supplements could be (ought to be) a substantial fraction of the* “*main*” *manuscript—this assessment is both a testament to the amount of work described in this paper, and a problem in that critical tests and examples of the method are buried*.

The previous version of the manuscript did not sufficiently describe the methods for data analysis and modeling, mainly because we were under the wrong impression that we needed to limit the length of the main text. Based on the reviewer suggestions (and our realization that *eLife* does not have an oppressive length limit), we have now reorganized the manuscript so that all methodological steps of data analysis and modeling—including newly presented controls (see below)—are now given in the main text. These moves include the following: Figure 3 now presents MOHCA-seq data for a model hairpin RNA construct (previously Figure 1—figure supplement 1), with newly added quantification of strand scission and reverse transcription stop rates, as well as a comparison of MOHCA-seq and MOHCA-gel datasets (previously Figure 1–figure supplement 3) and a comparison of different nucleotide attachment sites for radical sources (previously Figure 1–figure supplement 6). These figures are also described in the Results section in the main text, which is notably clearer and more complete as a result.

*Although the authors present many success stories, they do not describe the statistics on how many models are correct, how dependent the final models are on individual constraints (what happens when one leaves some out?), and so on. To help the reader judge the importance of the 3D MOHCA constraints, comparisons in the supplemental Tables 3 and 4 should be brought into the main text and possibly illustrated*.

We agree that the importance of MOHCA-seq constraints could not previously be assessed upon reading the main text. To better illustrate the importance of the MOHCA-seq constraints, we now: (i) have incorporated modeling statistics (with and without MOHCA-seq data) from Supplementary files 3 and 4 into Tables 1 and 2 in the main text, (ii) show the results of modeling with and without MOHCA-seq data in main text Figure 4 (previously Figure 2), (iii) have computed statistical significance values following the formulae of Hajdin, Weeks et al. and included these in Table 1 made several text updates, including:

In the subsection headed “MOHCA-seq tertiary proximity map complements M^2^ secondary structure for Rosetta modeling of P4-P6”: “Performing the same Rosetta modeling protocol without MOHCA-seq pairwise constraints led to a strikingly worse model with 38.3 Å accuracy to the crystal structure (Figure 4 and Table 1), demonstrating that MOHCA-seq constraints are required for accurate modeling.”

In the subsection “Multidimensional chemical mapping allows rapid modeling of diverse non-coding RNAs, including blind prediction challenges”: “Again, we tested the necessity of MOHCA-seq data for completing the MCM pipeline by performing identical modeling of all ncRNA domains without MOHCA-seq constraints (M^2^/Rosetta modeling); just as for the P4-P6 domain, these runs produced dramatically less accurate models, as can be visually assessed (Figure 4) and quantitated by both RMSD to the crystallographic structure and statistical significance (Hajdin et al., 2010) (Table 1).”

*A better discussion of the effects of noise, false distance constraints that arise from background cleavage of the RNA or other artifacts is needed*.

The effects of false distance constraints (MOHCA-seq hits that are inconsistent with the crystallographic structure in our benchmarks) were reported previously in our tables, which were provided as supplementary files, but they were not discussed thoroughly in the main text. We have added a better description of how noisy data are treated in signal processing:

Subsection “MOHCA-seq tertiary proximity map complements M^2^ secondary structure for Rosetta modeling of P4-P6”: “Additionally, the digital form of MOHCA-seq data enabled rigorous analysis to resolve further features. […] we apply a two-dimensional smoothing algorithm to produce the final proximity map for visualization (Figure 2—figure supplement 1).”

The false constraints appear reproducibly even after data averaging and this COHCOA analysis, suggesting that they are not due to noise. A remaining hypothesis, which we favor, is that these constraints are due to conformational fluctuations in the RNA structure and/or in the linkers. To better discuss effects of false constraints, we have included these fractions of unobeyed constraints in Table 2 and added discussion in the main text:

Subsection “MOHCA-seq tertiary proximity map complements M^2^ secondary structure for Rosetta modeling of P4-P6”: “For the P4-P6 benchmark case, we found that the cluster center of the MCM pipeline had an accuracy of 8.6 Å RMSD to the crystal structure (Figures 2 and 4 and Table 1). This suggests that MCM is capable of defining the global 3D structure of the RNA, without any crystallographic information, at near the resolution of a register shift (6 Å) even when weak MOHCA-seq features not explained by the crystal structure are used to constrain modeling.”

Subsection “Multidimensional chemical mapping allows rapid modeling of diverse non-coding RNAs, including blind prediction challenges”: “The lack of correlation between the fraction of satisfied MOHCA-seq pairwise constraints and the accuracy of the models (Table 2) […] as has been observed in prior macromolecule modeling based on pairwise information. (Das et al. 2008, Thompson et al. 2012, Bowers et al. 2000)”

In addition, we have added a significant amount of main text to discuss the possibility of long-distance hits due to linker blurring (see response below on correlative oxidative damage pairs) as well as conformational flexibility (see responses to reviewer 2 below).

*My main concerns have to do with the distribution of the MOHCA-Seq reads and their interpretation as* “*distance constraints*”*. In the main text (and supplement), the authors note that they observe additional sequence reads from arrest of RT at damaged (oxidized) nucleotides that lie between the radical source and the site of cleavage. First, the authors assume these* “*extra*” *stops arise from non-scissile base damage, but of course, they could arise from strand scission events. The authors write as if these are in different categories, but this distinction is meaningless as the reason for the RT stop can't be ascertained after the fact*.

We agree that our language and experimental assessment of the types and extent of oxidative damage were unclear, but are very important for establishing the proposed method. The reviewer’s comments have led us to include additional experimental controls that quantitated the extent and sequence dependence of damage (we had these in hand from our pilot work but did not include them previously). For example, we observed similar results from both capillary electrophoresis and sequencing analysis that the total amount of RT- stopping damage exceeds scissile backbone damage by at least 6-fold. These results are presented in the new Figure 3. Concomitantly we have expanded the main text to include more complete discussion of damage mechanisms, and references to the extensive work on radical damage on nucleic acids:

Subsection “Deep proximity maps of ncRNA 3D structure from single experiments”: “We expected reverse transcription to then stop at the nucleotide tethered to the bulky radical source at its C2´ position […] the 5´ and 3´ ends of the resulting cDNAs still reported pairs of sites that were proximal in the RNA structure, albeit at somewhat longer distances than source-to-cleavage pairs.”

*In*
[Disp-formula equ4]*, are the authors simply attempting to model the distribution of their data as arising from a component that depends on the frequency of tethered Fe(II)-EDTA complexes and a component that is independent of having tethered complexes? I presume this is the idea behind the COHCOA correlation analysis, which rightly ignores the distinction between* “*oxidative damage*” *and* “*cleavage*”*. This could be explained more forthrightly*.

The reviewer is correct on this point, and we have amended our language to make this clearer:

Subsection “Closure-based •OH COrrelation Analysis (COHCOA)”: “The effects of the radical cleavage result in a one-dimensional damage profile *R*_*i*_ (corresponding to uncorrelated oxidative damage) and cleavage profile *B*_*j*_ and a two-point correlation function *Q*_*ij*_ (corresponding to correlated oxidative damage or radical source locations) which encodes the desired proximity map”

*On the other hand, as the total dose of Fe(II)EDTA-dependent hydroxyl radical cleavage increases, the frequency of stops at a cleaved residue will equal or exceed RT stops at the site of the tether. In this limit, any information about the pairwise correlation between the site of the tethered reagent and the site of cleavage will be lost, undermining the premise of the experiment. At what limit does this occur, and does this tendency already account for the very uniform map before COHCOA analysis*?

The described effect of overmodification by radicals does not actually occur; we did not explain this well in the original paper. In our experiments, ascorbate-induced, spatially localized oxidative damage appears to be ‘self-limiting’; the correlated signal in MOHCA-seq datasets remains visible even after 30 minutes of incubation with ascorbate, which is three times our standard reaction time of 10 minutes, without a rising background. Based on separate electrophoresis experiments on the ascorbate-activated RNA and model constructs, the self-limitation appears to be due to the radical source cleaving itself off the RNA after some time. Although we previously described this mechanism in our MOHCA-gel paper, we neglected to discuss this in the original submission. We have added discussion of this point to the main text:

Subsection headed “MOHCA-seq tertiary proximity map complements M^2^ secondary structure for Rosetta modeling of P4-P6”: “To test the robustness of the protocol, we varied several experimental parameters […]. Capillary electrophoresis of ascorbate-treated end-labeled RNA supported this mechanism of self-limiting the reaction; see also ref (Das et al. 2008).”

*Even when cleavage events are sparse, I had trouble understanding how the authors turn the distribution of RT stops into reliable distance constraints. Backbone cleavage arising from a tethered source of hydroxyl radical can be interpreted as a (blurred) vector between two residues. By contrast, sequence reads that start and end at two damaged or cleaved sites may arise when both residues are close to the radical source. However, one cannot be sure how close the sites are to each other, especially as one does not know where the Fe-EDTA was tethered. Presumably the inconsistencies arising from such ambiguities in the distance constraints are sorted out during the modeling, but what evidence do the authors have that this is the case*?

The reviewer is correct: the location of a radical source that causes two correlated oxidative damage events that are both 3´ of the radical source’s location, one of which terminates a reverse transcription product that started at the other position, is not reported by the resulting cDNA. Such pairs of oxidative damage events are nevertheless still correlated by virtue of each being correlated to the radical source location, but their spatial distances will be blurred compared to direct source-scission events. It was a working hypothesis that these distances would still allow accurate global structure modeling; our original language did not make this clear. We have added discussion of this point to the text:

Subsection entitled “Deep proximity maps of ncRNA 3D structure from single experiments“: “Since these sites of reverse transcription termination marked locations that were also nearby in three dimensions to the radical source […] this significantly larger number of pairwise proximities would permit rapid ncRNA modeling with comparable accuracy to the prior method (Figure 3).”

As the reviewer indicates, we suspected that the inconsistencies in a set of many pairwise MOHCA-seq constraints would be sorted out by highly-sampled Rosetta modeling, as it was in the previous MOHCA-gel study. Here, we found that we were able to recover the 3D structures of our benchmark RNAs even though some MOHCA-seq constraints were not satisfied by our final models. We have added discussion of this point in the subsection “Multidimensional chemical mapping allows rapid modeling of diverse non-coding RNAs, including blind prediction challenges”, as quoted in our response to an earlier point.

*In the modeling section, it is stated that 60%-80% of constraints were satisfied by the best models; to what degree does this reflect the level of cleavage or specific types of non-random noise (see below)*?

As described in an above response, we observed that the level of correlated cleavage in the MOHCA-seq reaction is self-limiting, and as noted in our next responses, the non-random two-dimensional background arising from uncorrelated oxidative damage is removed by the COHCOA analysis and, for a few noise features, by averaging experimental replicates. So we believe the percentage of constraints satisfied is likely not related to this uncorrelated background but instead to alternative conformational states or inconsistent pairwise constraints arising from signal due to double oxidative damage events, as discussed in the previous response and in our revised text in “Multidimensional chemical mapping allows rapid modeling of diverse non-coding RNAs, including blind prediction challenges”.

[Disp-formula equ3]
*assumes a uniform rate of background cleavage* b_j_*. Of course, the background cleavage is almost certainly non-uniform. It is well known that certain residues will be more prone to hydrolysis (or RNase digestion) than others, and the resulting distribution of reads will appear similar to a genuine signal from a tethered Fe(II)EDTA*.

Our previous language describing a “uniform rate of background cleavage” was unclear. We have updated the language to clarify that the background cleavage *b*_*j*_ is not uniform along the sequence of an RNA molecule but is independent of and uncorrelated with any specific radical source:

In the subsection headed “General MOHCA-seq analysis framework”: “First, we assume that the cleavage fraction that allows ssDNA adapter ligation, qjs is composed of a ‘background’ rate of cleavage *b*_j_ that is independent of source *s* but can vary with nucleotide j (e.g., due to inline attack during the fragmentation reaction), and additional, correlated cleavage due to hydroxyl radical-induced strand scission events (the desired MOHCA-seq signal), parameterized by πjs:

Cleavage at j from source s=qjs=1−(1−bj)(1−πjs)≈bj+πjs_”_

*Is this type of non-random background addressed by using a mock-treated control? How is the information from the control incorporated into the data analysis? It would be great to see examples of data sets in which the iron was left out, or data sets on a long double-stranded RNA, and so on, to get an idea of how well the method works in these idealized or control examples*.

We hope that our updated language for the previous response clarifies that the COHCOA analysis is explicitly formulated to account for this type of non-random background.

In terms of controls, a standard part of the MOHCA- seq experiment is to carry out the full protocol on a sample that is not treated with ascorbate, in parallel with the ascorbate-treated (fragmented) sample; this is similar to a mock-treated control where Fe(III) is not loaded on the tethered ITCB-EDTA. This control is to ensure that signal that we observe in the ascorbate-treated samples does not occur independent of radical source activation, but we do not directly use the control data in our analysis because COHCOA accounts for such non-random background. A control sample treated with ascorbate but without tethered radical sources is shown in Figure 3—figure supplement 1, and a control sample with tethered radical sources but not treated with ascorbate is shown in Figure 3—figure supplement 2, and they are referenced in the legend for Figure 3.

*Finally, a drawback of the original MOHCA was that the experimentally observed constraints were all short range with respect to the RNA sequence. A cutoff was applied so that these short-range constraints did not overwhelm the model building. A similar strategy is presumably used here: how sensitive are the models to this cutoff*?

The cutoffs for determining MOHCA constraints were not clear in the original manuscript. Briefly, the large background for short sequence separations (the black ‘diagonals’ in presented maps) prevented use of MOHCA-seq constraints for short sequence separations. Further, given the accuracy of the newly available mutate-and-map protocol for inferring secondary structure, we sought to only use MOHCA-seq to determine tertiary proximities for nucleotide pairs that are far in secondary structure. The discussion of this constraint determination has been significantly expanded in the main text:

Subsection headed “MOHCA-seq tertiary proximity map complements M^2^ secondary structure for Rosetta modeling of P4-P6”: “First, to reduce the computational expense of sampling helices […] Therefore, we used all data, including the few weak features not explained by the crystal structure (pink circles, Figure 2).”

*And what is the practical limit on the size of the RNA that can be modeled by this method*?

We have noted the practical limit on the sizes of RNAs that can be addressed by the MCM pipeline in the Discussion:

“The experimental cost of MCM grows quadratically with the length of the transcript, limiting current analysis to domains of hundreds of nucleotides; however, >1000-nt transcripts may become feasible as sequencing costs continue to decrease.”

Reviewer #3:

*The determination of RNA structures has been challenging. While RNA molecules have evaded easy crystallization at a larger scale for a long time due to their highly charged backbone, structure predictions based sequence has gained large interest. Chemical probing of their structures has complemented these approaches. RNA molecules are comparably simple at the primary sequence and easily accessible to enzymatic and chemical manipulation followed by sequencing*.

*Cheng et al. present an experimental-computational multi-step pipeline that combines a mutate-and-map approach (M2) with a deep-sequencing based version of multiplexed hydroxyl radical cleavage analysis. The authors use this updated version in what they call a multidimensional chemical mapping (MCM) pipeline to determine a number of RNA structures. Some were known structures in the context of the RNA-puzzle challenge, in part done blindly. Using their pipeline Cheng et al. propose the structures of three riboswitches with and without their ligands and find that those two states are structurally much more similar than previously thought. Attempts to crystallize those switches without ligands may force the molecules into an undesired stable conformation and be non-physiological or these conformations might never result in well diffracting crystals. Lastly they propose a first structure for the human HoxA9D IRES, a cellular IRES implicated in the translational regulation of developmental patterning*.

*The authors describe an interesting experimental pipeline that may move the determination of RNA structures out of the hands of few specialized RNA biophysics labs into the hands of molecular biologists with access to deep sequencing equipment. The experimental method is relatively straightforward for anybody who has generated deep-sequencing libraries. Even though far from trivial, the adaptation of the protocol can provide a handle to address structural features at a larger scale which is very desirable in different fields of RNA biology*.

*The method is a further step in structure determination, but has only been verified on a very limited set of RNA structures. This makes some of the predictions difficult to assess*.

We wish to note that the presented benchmark, with 6 ‘benchmark’ RNAs of known structure and 4 cases with unknown (but interesting) structure, is at a larger scale than prior studies introducing new experimental methods (including our own MOHCA-gel paper and recent 3D approaches based on NMR). It also includes a blind test, which provides rigorous assessment, and is unusual in modeling.

*As an example:*
Figure 2
*shows the refinement of P2.1 towards P4-P6. Other structural elements are not refined at all (purple loop). Why is that? Is a global RMSD score the best measure to describe the refinement of the structure? A wrong prediction of functionally critical residues may hamper the use of the method significantly. Can the authors use their data and compare whether the structural refinements are meaningful by focusing on the critical elements for their direct comparison*?

We agree with the reviewer that a global RMSD score is not the best representation of the refinement of the lariat-capping ribozyme model, and we have thus added the RMSDs over the refined elements (P2.1 and P4-P6) from MCM and M^2^/Rosetta modeling to the figure legend (Figure 4), in addition to stating their modeling results and statistics in Tables 1 and 2.

*Due to the linearity of their sequencing approach the method can only detect cleavage that occurs 3' of the •OH source. The authors themselves suggest, that circularization of the RNA might circumvent this issue but argue against it. This assumption may have major consequences on the quality of the prediction*.

We have removed this text, as it was confusing. Circularization will not be generally applicable to any RNA domain, but may enhance the information content for cases that can be circularized; we are testing this procedure now but the results are beyond the scope of this study.

*The manuscript is to a large extent the description of a new method. It is absolutely essential that the protocol is immediately applicable to different questions by different labs. In their Material and methods the authors are not consistent with the information they provide. It would be important if they provide information on how much material they recover at each step of the protocol (Post OH step, 3'end repair, ligation…) as well as the reaction volumes of all steps (Results)*.

We thank the reviewer for pointing out these omissions. We have added these details to the Materials and methods section, including estimated yields for each step.

*For the description of the riboswitch structures, I prefer to see the apo and the bound structure overlaid to form a better opinion on the extent of structural rearrangement of the two*.

We now show the ligand-bound crystallographic structures next to the MCM models of unbound riboswitches, in the same orientation, in the figures depicting these comparisons (Figures 5, 6 and 7). We tried overlaying the structures but found that it was difficult to make the visual comparison.